# Genetic and phylogenetic uncoupling of structure and function in human transmodal cortex

Sofie L. Valk [1,2,3 ✉], Ting Xu [4], Casey Paquola[5,6], Bo-yong Park [5,7,8], Richard A. I. Bethlehem[9], Reinder Vos de Wael[5], Jessica Royer [5], Shahrzad Kharabian Masouleh [2,3], Şeyma Bayrak[1], Peter Kochunov[10], B. T. Thomas Yeo[11,12,13,14,15], Daniel Margulies [16], Jonathan Smallwood[17], Simon B. Eickhoff[2,3,18] & Boris C. Bernhardt [5,18]

Brain structure scaffolds intrinsic function, supporting cognition and ultimately behavioral flexibility. However, it remains unclear how a static, genetically controlled architecture supports flexible cognition and behavior. Here, we synthesize genetic, phylogenetic and cognitive analyses to understand how the macroscale organization of structure-function coupling across the cortex can inform its role in cognition. In humans, structure-function coupling was highest in regions of unimodal cortex and lowest in transmodal cortex, a pattern that was mirrored by a reduced alignment with heritable connectivity profiles. Structure-function uncoupling in macaques had a similar spatial distribution, but we observed an increased coupling between structure and function in association cortices relative to humans. Meta-analysis suggested regions with the least genetic control (low heritable correspondence and different across primates) are linked to social-cognition and autobiographical memory. Our findings suggest that genetic and evolutionary uncoupling of structure and function in different transmodal systems may support the emergence of complex forms of cognition.

[1] Otto Hahn Group Cognitive Neurogenetics, Max Planck Institute for Human Cognitive and Brain Sciences, Leipzig, Germany. [2] Institute of Neuroscience and Medicine, Brain & Behaviour (INM-7), Research Centre Jülich, FZ Jülich, Jülich, Germany. [3] Institute of Systems Neuroscience, Medical Faculty, Heinrich Heine University Düsseldorf, Düsseldorf, Germany. [4] Center for the Developing Brain, Child Mind Institute, New York, NY, USA. [5] Multimodal Imaging and Connectome Analysis Lab, McConnell Brain Imaging Centre, Montreal Neurological Institute and Hospital, McGill University, Montreal, QC, Canada. [6] Institute of Neuroscience and Medicine, Structural and functional organisation of the brain (INM-1), Research Centre Jülich, Jülich, Germany, FZ Jülich, Jülich, Germany. [7] Department of Data Science, Inha University, Incheon, South Korea. [8] Center for Neuroscience Imaging Research, Institute for Basic Science, Suwon, South Korea. [9] Department of Psychiatry, Cambridge University, Cambridge, UK. [10] Department of Psychiatry, University of Maryland School of Medicine, Baltimore, MD, USA. [11] Department of Electrical and Computer Engineering, National University of Singapore, Singapore, Singapore. [12] Centre for Sleep and Cognition (CSC) & Centre for Translational Magnetic Resonance Research (TMR), National University of Singapore, Singapore, Singapore. [13] N.1 Institute for Health & Institute for Digital Medicine (WisDM), National University of Singapore, Singapore, Singapore. [14] Martinos Center for Biomedical Imaging, Massachusetts General Hospital, Charlestown, MA, USA. [15] Integrative Sciences and Engineering Programme (ISEP), National University of Singapore, Singapore, Singapore. [16] Neuroanatomy and Connectivity Lab, Institut de Cerveau et de la Moelle epiniere, Paris, France. [17] Department of Psychology, Queen's University, Kingston, ON, Canada. [18] These authors contributed equally: Simon B. Eickhoff, Boris C. Bernhardt. ✉email: s.valk@fz-juelich.de

Cognition helps an animal to satisfy core biological goals in a changing environmental context. In humans, cognition allows our species to successfully navigate through a broad array of situations and socio-cultural contexts. Although the need for flexible cognition is well-established, it remains unclear how a relatively static brain organization can give rise to functional patterns with sufficient flexibility to encounter and manage complex and culturally rich landscapes found in human societies.

Contemporary perspectives suggest that higher human cognition is grounded in a cortical organization that encompasses parallel axes of microstructural differentiation and function[1] [for nomenclature: Table 1]. On the one hand, sensory/motor systems as well as unimodal association cortices are involved in operations related to perceiving and acting in the outside world. These systems are differentiated from transmodal systems that are less tied to a specific modality, and are increasingly engaged in abstract and self-generated cognition, and interface with the "internal milieu"[1–4]. These functional differences are reflected in well-established differences in the microstructure of sensory/motor and transmodal cortex. Histological studies demonstrate that unimodal sensory and motor regions have more distinctive lamination patterns relative to agranular/dysgranular transmodal cortex with less apparent lamination[5]. Complementing these findings, in vivo studies have shown that transmodal regions have overall lower myelin content[6–9], yet more complex dendritic arborization patterns, which could facilitate integrative processing and increased potential for plasticity[10]. According to its classic definition[1], transmodal cortex encompasses both paralimbic and heteromodal association networks[11], and in particular the default mode and fronto-parietal functional networks that are particularly expanded in humans[11]. These latter two networks are known to participate in a broad class of abstract cognitive processes[12], including autobiographical memory[13,14], language[15–17], as well as executive control[2,18,19].

Post-mortem studies in non-human animals together with emerging data in humans[20–22] suggest that regions with a similar cytoarchitecture are also more likely to be structurally and functional interconnected, an observation framed as the "structural model" of brain connectivity[23,24]. Yet, it remains to be established how mappings in cortical structure and function vary across different cortical areas. Recent in vivo work suggests that structure–function coupling as measured by the association of white matter tractography and functional connectivity is progressively diminished towards transmodal cortex relative to sensory/motor and unimodal areas[25–27]. Similar findings can also be observed when studying associations between cortical microstructure based on T1-weighted/T2-weighted (T1w/T2w) and functional connectivity[22], collectively pointing to a differential organization of transmodal systems in terms of cortical structure and function relative to unimodal systems[22,28,29].

Transmodal systems are assumed to play a role in more abstract cognition that is less constrained to specific modalities of information, so a reduction in the constraining influence of cortical structure on function in transmodal systems may be an important evolutionary adaptation supporting human cognition[1,3,11,27]. Heteromodal regions have been reported to show increased expansion in surface area and untethering from external and internal inputs, associated with non-hierarchical circuit properties[1,11]. Such untethering may enable parallel and recursive computations linked to human cognition[1,11]. To better understand functional properties of these untethered regions of cortex, our study set out to understand if associations between cortical structure and function may enable an architecture hypothesized to give rise to abstract human cognition[11,12] using two related approaches. First, we examined the heritability of structure–function relationships across the cortex in humans. Heritability serves as a backbone for evolutionary change, as natural selection acts upon inherited traits under variation[30]. In humans, patterns of cortical microstructure and functional connectivity are heritable, indicating partial genetic control over individual variation[31–37] and our study aimed to understand if this relationship extends to transmodal cortex where function is thought to be untethered. Complementing the heritability assessment in humans, we examined how structure–function associations seen in our species are preserved in non-human primates (NHP). Phylogenetic comparisons between humans and NHP can help establish the degree to which specific brain patterns are conserved across primate species[38,39] providing a mechanistic perspective on human cognition that complements work on heritability. Previous work has shown that spatial

**Table 1 Replication of table in Paquola, 2019 of Mesulam classes nomenclature.**

| Mesulam, 2000 | Regional names[a] | Brodmann, 1909 | Von Economo and Koskinas, 1925 |
|---|---|---|---|
| Idiotypic primary | Striate | 17 | OB, OC |
| | Auditory | 41, 42 | TC |
| | Somatosensory | 3a, 3b, 1, 2 | PA, PB, PC |
| | Motor | 4, 6 | FA |
| Modality-specific unimodal | Upstream peristriate | 18, 19 | OA |
| | Inferotemporal | 20, 21, 37 | PH, TE |
| | Superior temporal | 22 | TB, TD |
| | Superior parietal lobule | 5, anterior 7 | Part of PE |
| | | | Part of PF |
| | Inferior parietal lobule | Anterior 40 | FB |
| | Premotor | Anterior 6, posterior 8, 44 | FCBm |
| Higher-order heteromodal | Prefrontal cortex | 9, 10, 45, 45, 47, anterior 11, anterior 12, anterior 8 | FC, FD, FDdelta, FDT, FE |
| | Posterior parietal | Posterior 7, 39, 40 | PD, PG, parts of PE & PF |
| | Lateral temporal | Parts of 21 and 37 | Part TE |
| | Parahippocampal | Parts of 36 and 37 | TF |
| Paralimbic | Orbitofrontal cortex | Posterior 11, posterior 12, 13 | FF, FG, FH, FJK, FLMH |
| | Insula | 14–16, | IA, IB |
| | Temporal pole | 38 | TG |
| | Parahippocampal | 27,28,35 | HA, HB, HC |
| | Cingulate | 23–26, 29–33 | LA1, LA2, LC1–3, LD, LE |

[a]Ascribed by Mesulam (2000). The transmodal cortex is defined as areas without modality-specific input, i.e., heteromodal and paralimbic isocortex (based on Mesulam, 2013).

variations in cortical microstructure[15,40] and functional connectivity seen in humans[41,42] are present in NHP, but that several evolutionary changes may have emerged in humans. Comparing markers of cortical myelin content between NHP and humans, recent studies have established that the arcuate fasciculus, a fiber bundle that connects temporal and parietal cortex, underwent evolutionary modifications particularly in humans[15]. More generally, humans have a greater proportion of transmodal cortex than NHP[40,43], a pattern of cortical expansions that may have also altered the locations of heteromodal functional networks when comparing humans and macaques[41].

Combining heritability and cross-species approaches to probe structure–function associations, our study examined genetic processes that determine the uniqueness of human structure–function coupling and decoupling in transmodal areas, which are assumed to play key roles in abstract human cognition[11,12]. In particular, we quantified node-level correlation of T1w/T2w microstructure profile covariance (MPC)[22] and resting-state functional connectivity (rsFC). The T1w/T2w ratio has been shown to reflect myelin content[44–47] but also iron, water, as well as cytological variations including dendritic arborization, cell size, and cell density[8,48,49]. To study heritability, we utilized the pedigree design and multimodal imaging data of the Human Connectome Project (HCP) S1200 release[50]. Equivalent analyses in macaques were performed using the PRIMate-Data Exchange repository[38], which allowed us to examine phylogenetic differences in microstructural and functional organization. We combined node-level network neuroscience approaches with the use of unsupervised dimensionality reduction techniques, which identified large-scale microstructural and functional gradients and provided a coordinate system to map genetic and evolutionary influences on cortical organization[29,51,52]. In particular, we evaluated the relationship between structure–function coupling with evolutionary axes functional reorganization[41], and the dual origin model[53,54]. The latter model assumes that cortical areas develop from waves of laminar differentiation that have their origin in either the piriform cortex (paleocortex) or the hippocampus (archicortex)[53,55]. Finally, we contextualized the likely functional profile of these regions through meta-analytical data from the task-based functional magnetic resonance imaging (fMRI) literature[56]. We also performed various robustness analyses to assess the stability of our findings. See Supplementary Fig. 1 for schematic of analyses.

In this work, we show that structure–function coupling (i.e., the regional correlation between microstructural profile covariance and resting-state functional connectivity) in humans is highest in primary regions and lowest in transmodal cortex. Uncoupling of structure and function is paralleled by genetic uncoupling, as probed by twin-based heritability analysis, particularly in heteromodal regions. Structure–function uncoupling in macaques has overall a comparable spatial distribution to that seen in humans. However, in heteromodal regions, structure and function are more coupled in macaques than in humans. Structure–function patterns were confirmed when assessing organizational gradients of microstructure and function. Our findings suggest genetic and phylogenetic uncoupling of structure and function in transmodal systems believed to play important roles in human cognition.

## Results
### Cortex wide decoupling of function and structure (Fig. 1). We first established the spatial distribution of structure–function coupling in the human brain, using node-level association analyses. Specifically, we mapped how patterns of intrinsic functional connectivity reflect similarity of cortical microstructure across all cortical regions, using the S1200 sample of the Human Connectome Project young adult dataset[50] (see Methods for details on participants and neuroimaging processing). To construct microstructure profile covariance (MPC) matrices, we sampled intracortical T1w/T2w values at 12 different cortical depths[22], and correlated dept-wise cortical profiles between the parcels (Fig. 1A). To control for curvature effects, equivolumetric surfaces were used[57]. Resting-state functional connectivity (rsFC) matrices were calculated by cross-correlating the neural time-series between all pairs of 400 cortical nodes[58] (Fig. 1B). Correlating node-wise patterns in both measures, averaged across participants (Fig. 1C), we observed an overall edge-level association between MPC and rsFC, in line with the predictions of the structural model[21]. As expected, however, there was also a progressively decreasing correspondence between MPC and rsFC ($r = 0.525$, $p < 0.001$) along a sensory-fugal gradient of cytoarchitectural classes, capturing cytoarchitectural complexity [ref. [59], for nomenclature: Table 1 and Fig. 1C] from high correlations in primary regions (mean ± SD: Spearman's $r = 0.465 ± 0.181$), to weak correlations unimodal association cortices (Spearman's $r = 0.149 ± 0.263$), followed by close to zero correlations in transmodal cortex (heteromodal association cortices, Spearman's $r = 0.063 ± 0.199$; paralimbic areas, Spearman's $r = 0.012 ± 0.146$). Findings were consistent in a replication sample ($N = 50$)[60], when using an alternative node parcellation (Supplementary Results). Relative to sensory/motor and unimodal areas, transmodal systems show the strongest decoupling between cortical microstructure and functional connectivity.

### Genetic control over structural and functional connectivity profiles (Fig. 2). Having documented reductions in the association between in MPC and rsFC in transmodal cortices, we next examined whether this difference is heritable, i.e., under genetic control. The HCP S1200 sample contains both unrelated as well as genetically related individuals, allowing us to analyze heritability through maximum likelihood analysis. We first computed the node-wise heritability of MPC and rsFC using Sequential Oligogenic Linkage Analysis Routines (http://www.solar-eclipse-genetics.org; Solar Eclipse 8.4.0.). To assess to what extent rsFC and MPC interregional patterns were under genetic control, we compared mean seed-wise connectivity/covariance profiles with seed-wise heritability of each measure separately. This index provides us with a local measure to what extent mean and heritable patterns are similar, analogous to the MPC-rsFC coupling measure. In this context, high correlations suggest that edges with high MPC/rsFC correspondence also show high heritability, whereas low correlations suggest low heritability. At the whole network level, MPC was heritable (mean ± SD h2 = 0.167 ± 0.030), but effects appeared weaker than for rsFC (h2 = 0.340 ± 0.042). At a node level, primary sensory/motor regions showed positive correlation between mean and heritable connectivity profiles for both MPC and rsFC, whereas transmodal regions showed less correlations between mean and heritable connectivity profiles in both measures (MPC: mean ± SD primary (idiotypic): Spearman's $r = 0.482 ± 0.145$; unimodal: 0.318 ± 0.220; heteromodal: 0.168 ± 0.191; paralimbic: 0.295 ± 0.206 and rsFC: primary: Spearman's $r = 0.656 ± 0.185$; unimodal: 0.537 ± 0.204; heteromodal: 0.385 ± 0.210; paralimbic: 0.446 ± 0.172). Assessing differences in genetic coupling as a function of cytoarchitectural class, we found that for MPC, only heteromodal regions showed consistently reduced coupling relative to other classes (idiotypic: $t = -12.363$, $p < 0.001$; unimodal: $t = -6.129$, $p = 0.006$; paralimbic: $t = -4.022$, $p = 0.03$). For rsFC transmodal classes

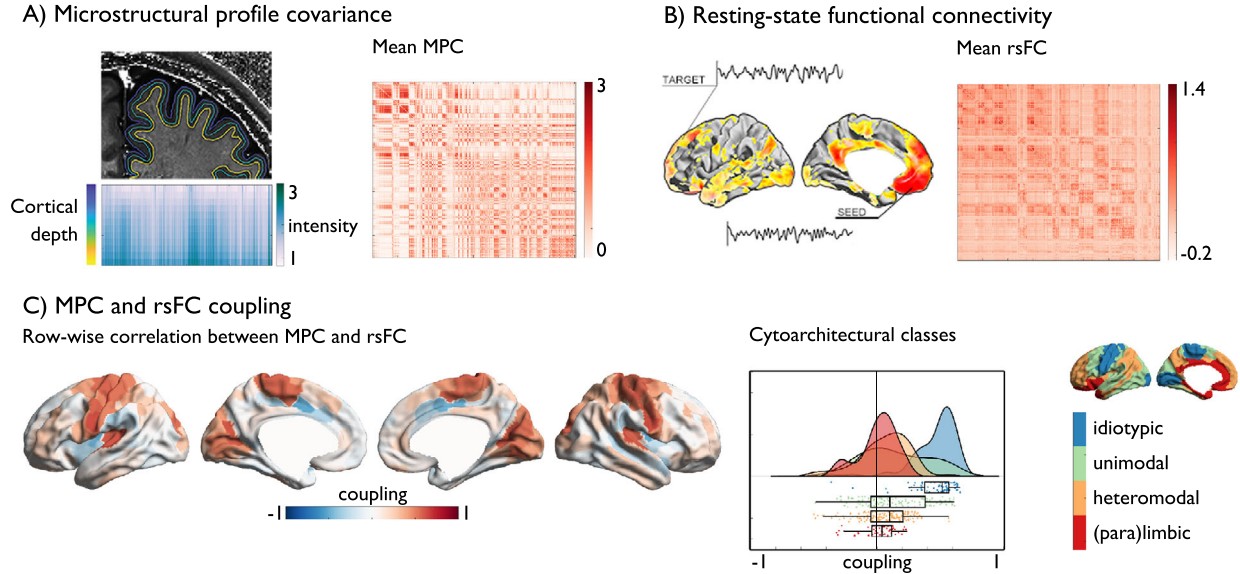

**Fig. 1 Structure–function coupling and heritability in human cortical regions. A** Microstructural profile covariance (MPC) was chosen to map networks of microstructural similarity for each cortical node, sorted along cytoarchitectural class[22,59]; **B** Resting-state functional connectivity (rsFC) analysis maps nodal patterns of intrinsic functional connectivity, sorted along cytoarchitectural class; **C** Row-wise coupling of MPC and rsFC Middle: raincloud plot of distribution within cytoarchitectural classes of coupling in the 400 Schaefer parcels, box shows the median and interquartile (25–75%) range, whiskers depict the 1.5*interquartile range (IQR) from the quartiles; Right: Reference visualization cytoarchitectural class. Source data are provided as a Source Data file.

displayed reductions in genetic coupling relative to non-transmodal classes (heteromodal: idiotypic: $t = -9.320$, $p < 0.001$; unimodal: $t = -6.062$, $p < 0.001$ and paralimbic regions (idiotypic: $t = -6.340$, $p < 0.001$; unimodal: $t = -2.780$, $p = 0.024$). Genetic coupling between heteromodal and paralimbic regions showed no significant difference in rsFC ($t = 1.839$, $p_{uncorr} = 0.067$). These associations were not significantly linked to test–retest reliability affecting the measures in the same direction and showed comparable patterns in an alternative measure of heritability of rsFC that accounts for intra-subject variations[61] (Supplementary Results). In addition, we assessed the heritability of MPC-rsFC coupling. Overall coupling was heritable ($h2 = 0.260 \pm 0.088$), at the same time, we found that in particular coupling in heteromodal regions was less heritable relative to other cortical types (primary (idiotypic): $h2 = 0.284 \pm 0.073$; unimodal: $0.270 \pm 0.093$; heteromodal:$0.235 \pm 0.087$; paralimbic: $0.273 \pm 0.082$) with only differences between heteromodal and other cytoarchitectural class being below $p < 0.05$ (idiotypic: $t = -4.158$, $p < 0.001$; unimodal: $t = -3.223$, $p = 0.006$; paralimbic: $t = -2.702$, $p = 0.03$), (Fig. 2). This indicates that while in primary regions the correlation between MPC and rsFC as well as the correspondence of mean and heritable connectivity profiles is strong, we observed genetic decoupling in transmodal regions, particularly in heteromodal areas.

**Correspondence between microstructure and function in non-human primates (Figs. 3 and 4).** Having established that structure–function coupling is reduced in human transmodal cortex, we next examined whether the above structure–function relations are also seen in NHPs, by evaluating 19 macaques from the PRIMate-Data Exchange who had microstructural MRI and resting-state fMRI available[38]. Analogous to the human analysis, we created MPCs using multiple equivolumetric surfaces between pial and gray matter/ white matter surfaces to extract depth-

dependent T1w/T2w profiles in each monkey (details in Supplementary Results). We carried out an rsFC analysis, and co-registered humans and macaques using a recently introduced cross-species alignment[41]. Comparing edges of MPC and rsFC in macaques, we found an overall correlation between MPC and rsFC ($r = 0.22$, $p < 0.0001$). Correspondence between MPC and rsFC was similar in both species (Spearman's $r = 0.48$, $p_{spin} = 0.016$), albeit stronger in macaques ($t = 7.020$, $p < 0.001$). Comparing associations at the microstructural level, using cytoarchitectural classes (Table 1), we found no difference between coupling in paralimbic regions (uncoupled in humans and macaques; $t = 0.854$, $p > 0.1$) and primary sensory/motor regions (coupled in humans and macaques; $t = 0.063$, $p > 0.1$). However, both unimodal ($t = 6.399$, $p < 0.001$) and heteromodal association regions ($t = 6.445$, $p < 0.001$) were generally more coupled in macaques than in humans. Replacing the macaque rsFC matrix with two other samples, including males and females with a different age-range, and awake as well as anesthetized monkeys, yielded broadly similar patterns of structure–function coupling (Fig. 4 and Table 2). Differences between humans and macaques did not show a significant relation with differences in tSNR. Moreover, structure–function coupling in humans was reproducible in another human sample (eNKI; $N = 100$, age = 18–40 years) preprocessed analogous to the macaque sample ($r = 0.94$ with structure–function coupling in HCP, difference between humans and macaques as a function of cytoarchitectural class: idiotypic $t = -1.199$, $p > 0.1$; unimodal $t = 5.161$, $p < 0.001$; heteromodal $t = 5.690$, $p < 0.001$; paralimbic $t = -2.182$, $p > 0.1$), suggesting differences between humans and macaques went beyond differences in preprocessing (Supplementary Results). This analysis shows that while humans and macaques have broadly similar spatial trends of reduced structure–function correspondence from sensory to transmodal areas, both uni- and heteromodal association cortices are characterized by reductions in structure–function coupling in humans compared to NHPs.

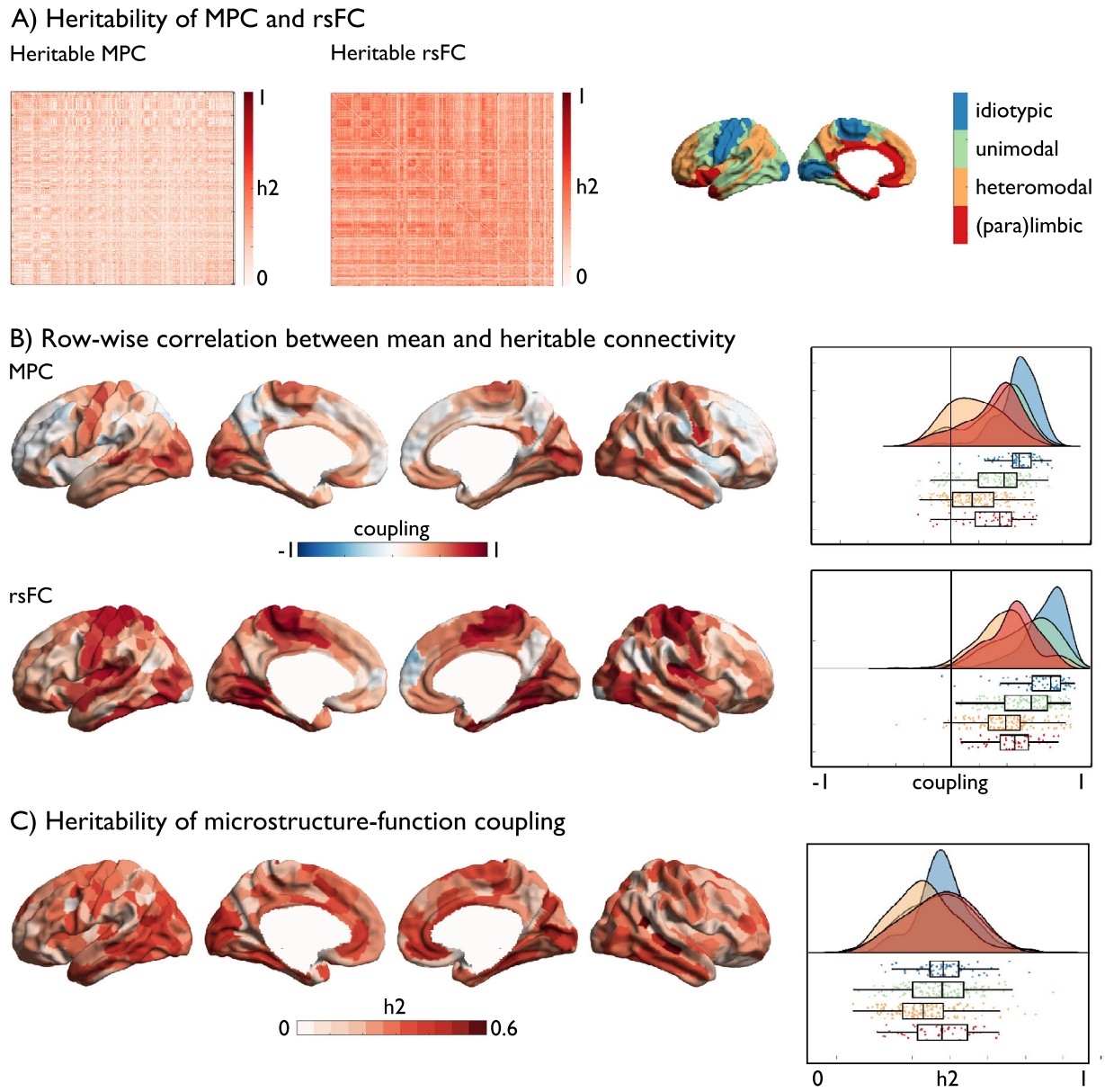

**Fig. 2 Genetic basis of structure–function coupling. A** Heritability of MPC and rsFC; **B** Row-wise association between mean and heritable seed-wise connectivity—reflecting genetic control over connectivity profiles, Right: distribution of coupling in 400 parcels per cytoarchitectural class, box shows the median and interquartile (25–75%) range, whiskers depict the 1.5*IQR from the quartile; **C** Heritability of microstructure-function coupling, Right: distribution of heritability in 400 parcels per cytoarchitectural class, box as in **B**[22,59]. Source data are provided as a Source Data file.

**Organizational differences in microstructure and functional connectivity (Figs. 5 and 6).** To further understand divergence between MPC and rsFC, we evaluated the differences in main organizational axes of MPC/rsFC. Applying non-linear dimensionality reduction techniques to both modalities separately[62,63], we observed that both MPC and rsFC followed similar spatial axes as their heritability (Fig. 5). Next, we calculated the nodal difference between standardized and aligned MPC and rsFC principal gradients ($\Delta_{MPC-rsFCG1}$). In line with previous reports[22], gradients were uncoupled in heteromodal regions ($t = -3.878$, $p < 0.01$), but not in other classes (idiotypic: $t = -2.42$, $p > 0.06$, unimodal: $t = -1.731$, $p > 0.1$, and paralimbic: $t = 2.51$, $p > 0.05$). The MPC gradient ran from sensory/motor to paralimbic regions, while the rsFC gradient radiated from sensory/motor to

heteromodal networks (Fig. 6A). Organizational gradients did not correlate with test–retest intraclass correlations of the respective measures (Supplementary Results), suggesting that differences between rsFC and MPC G1 are not linked to variable levels of noise. We then selected homolog gradients of MPC and rsFC in macaques (Supplementary results) and again observed the most marked difference within heteromodal networks ($t = -5.557$, $p < 0.001$), but no other classes (idiotypic: $t = 0.554$, $p > 0.1$, unimodal: $t = -2.24$, $p > 0.1$, paralimbic: $t = 1.925$, $p > 0.1$). There was also a moderate positive correlation between MPC and rsFC gradient differences in humans and macaques (Spearman's $r = 0.176$, $p < 0.01$, $p_{spin} > 0.1$), and no significant difference in loadings of the different cytoarchitectural classes (idiotypic: $t = -1.388$, $p > 0.1$; unimodal: $t = -0.357$, $p > 0.1$; heteromodal:

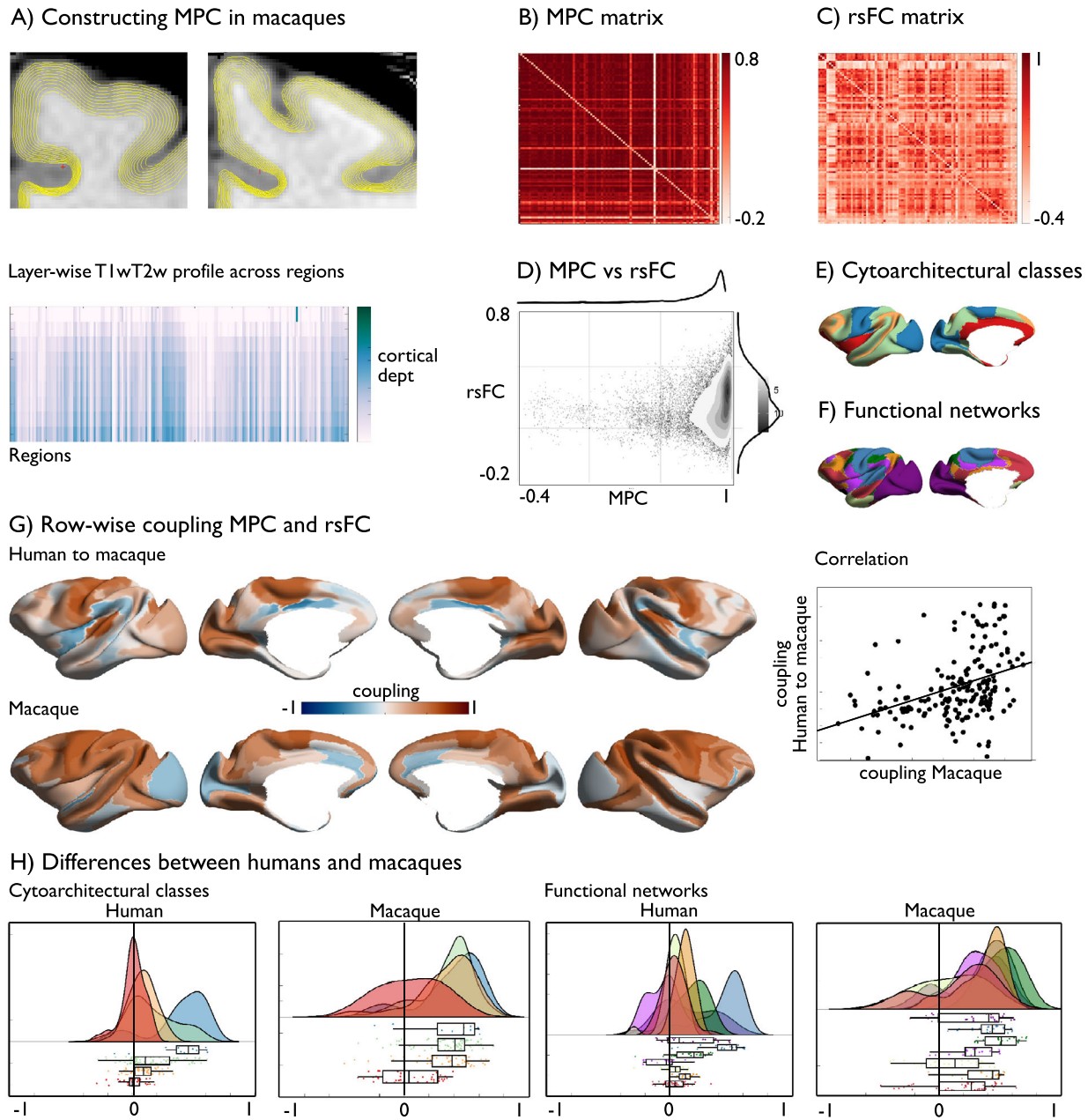

**Fig. 3 Microstructure-function coupling in macaques. A** Creating MPC in macaques; **B** MPC matrix in macaques, ordered along cytoarchitectural class based on ref. [59] and Markov labels[116]; **C** rsFC matrix in macaques, ordered along cytoarchitectural class; **D** Correspondence between MPC and rsFC in macaques; **E** Cytoarchitectural classes; **F** Functional communities based on ref. [28]; **G** Row-wise association of MPC and rsFC; upper panel: human map in macaque space; lower panel: macaque map; right: scatter between human and macaque MPC-rsFC coupling; **H** Raincloud plots of coupling in humans and macaques as a function of cytoarchitectural class and functional communities[28] in macaque space (182 parcels of Markov parcellation), boxes show the median and interquartile (25–75%) range, whiskers depict the 1.5*IQR from the quartile. Source data are provided as a Source Data file.

$t = 1.942$, $p > 0.1$; paralimbic: $t = -1.63$, $p > 0.1$) across species. This suggests that the differentiability of axes organizing transmodal regions for MPC and rsFC is present in both humans and macaques, establishing it as a general feature of the primate cortex.

**Multiscale quadrants of structure–function coupling (Fig. 7).** The axes of structure–function coupling and gradient differences offer a two-dimensional coordinate system to visualize and

conceptualize how cortical organization is linked to function. As expected, the axis describing structure–function coupling differentiates primary from transmodal regions (upper and lower half of the quadrant). Conversely, the MPC-rsFC gradient difference axis segregates heteromodal and paralimbic regions. To establish whether this space has implications for functional organization, we assessed its ability to differentiate different motifs of neural structure, evolution, and function (see Methods).

We first projected cytoarchitectural classes and intrinsic functional communities into our two-dimensional coordinate

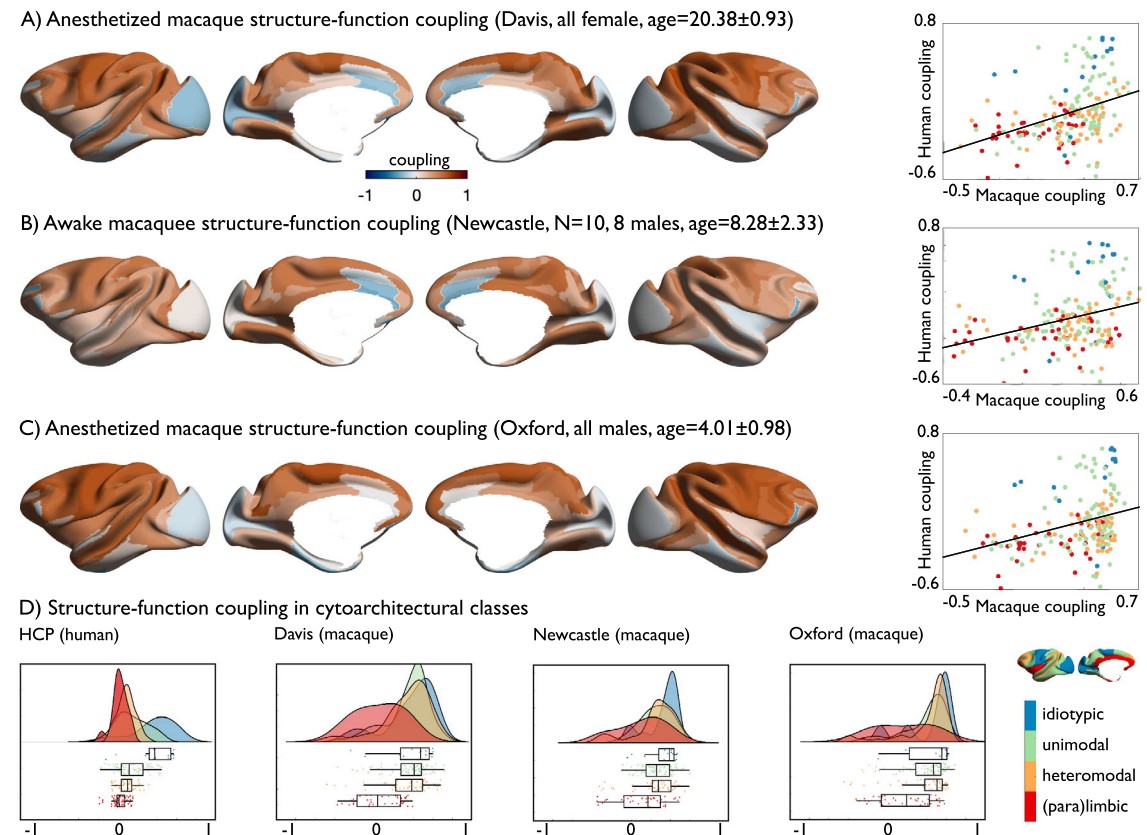

**Fig. 4 Structure–function coupling in different macaque samples. A** Anesthetized macaque (sample: Davis) structure–function coupling and correlation with human structure–function coupling projected to macaque space; **B** Awake macaque (sample Newcastle) structure–function coupling and correlation with human structure–function coupling projected to macaque space; **C** Anesthetized macaque (sample Oxford) structure–function coupling and correlation with human structure–function coupling projected to macaque space; **D** Structure–function coupling averaged in cytoarchitectural classes[59] across 182 parcels of the Markov parcellation for HCP, Davis, Newcastle and Oxford samples, boxes show the median and interquartile (25–75%) range, whiskers depict the 1.5*IQR from the quartile. Source data are provided as a Source Data file.

| Table 2 Difference in structure–function coupling between humans and macaque across samples. | | | | |
| --- | --- | --- | --- | --- |
| **Sample** | **Idiotypic [mean (SD)]** | **Unimodal** | **Heteromodal** | **Paralimbic** |
| Davis- HCP | 0.0628 ($p > 0.1$) | 6.399 ($p < 0.001$) | 6.445 ($p < 0.001$) | 0.063 ($p > 0.1$) |
| Newcastle-HCP | −0.6713 ($p > 0.1$) | 4.516 ($p < 0.001$) | 4.759 ($p < 0.001$) | 2.184 ($p > 0.1$) |
| Oxford-HCP | 0.699 ($p > 0.1$) | 7.744 ($p < 0.001$) | 7.905 ($p < 0.001$) | 3.157 ($p = 0.01$) |

Difference between human, HCP, sample as a function of cytoarchitectural class as measured by a two-sample $t$-test, reported $p$-values corrected for number of classes (4).

system to understand the large-scale communities linked to reduced structural–function coupling (Fig. 7A). In humans and macaques, both upper quadrants include sensory/motor and unimodal networks. However, while in macaques the left upper quadrant is also occupied by fronto-parietal and default mode networks, these networks are in the lower left quadrant in humans. The lower right quadrant is dominated by the limbic and ventral attention networks in humans, and the limbic network in macaques. Comparing the relative number of nodes distributed in each quadrant, more areas were uncoupled in humans relative to macaques ($\chi^2$: 16.044, $p < 0.001$), particularly in the left half of the quadrant ($\chi^2$: 10.249, $p = 0.0014$).

In line with the observed cross-species differences, the quadrants also reflected a differentiation of cortical reorganization between humans and macaques reported in prior findings[41], and these differences may reflect archi- vs paleocortical trends hypothesized by the dual origin model of cortical differentiation[53]

(Fig. 7B). Furthermore, projecting gene expression maps from the Allen Human Brain Atlas (AHBA) into the 2D space[64] suggested that the quadrants may also capture different biological pathways. For example, the lower left quadrant was associated with genes that are mainly expressed in prenatal states in cortical areas, while the lower right quadrant was additionally associated with both the prenatal expression of nodes in the hippocampus together with the cortical regions and thalamus, striatum, and hippocampus postnatally (Fig. 7C and Table S2–5). Thus, the genetic uncoupling observed in twins in our main analyses may reflect differential time-windows of developmental expression in cortical and non-cortical regions.

Finally, we identified the most likely cognitive consequences of the structure–function decoupling in transmodal cortex. To this end, we mapped cognitive ontologies based on NeuroSynth into the two-dimensional space[56] (Fig. 7D). Quadrants with high structure–function coupling included primary and unimodal

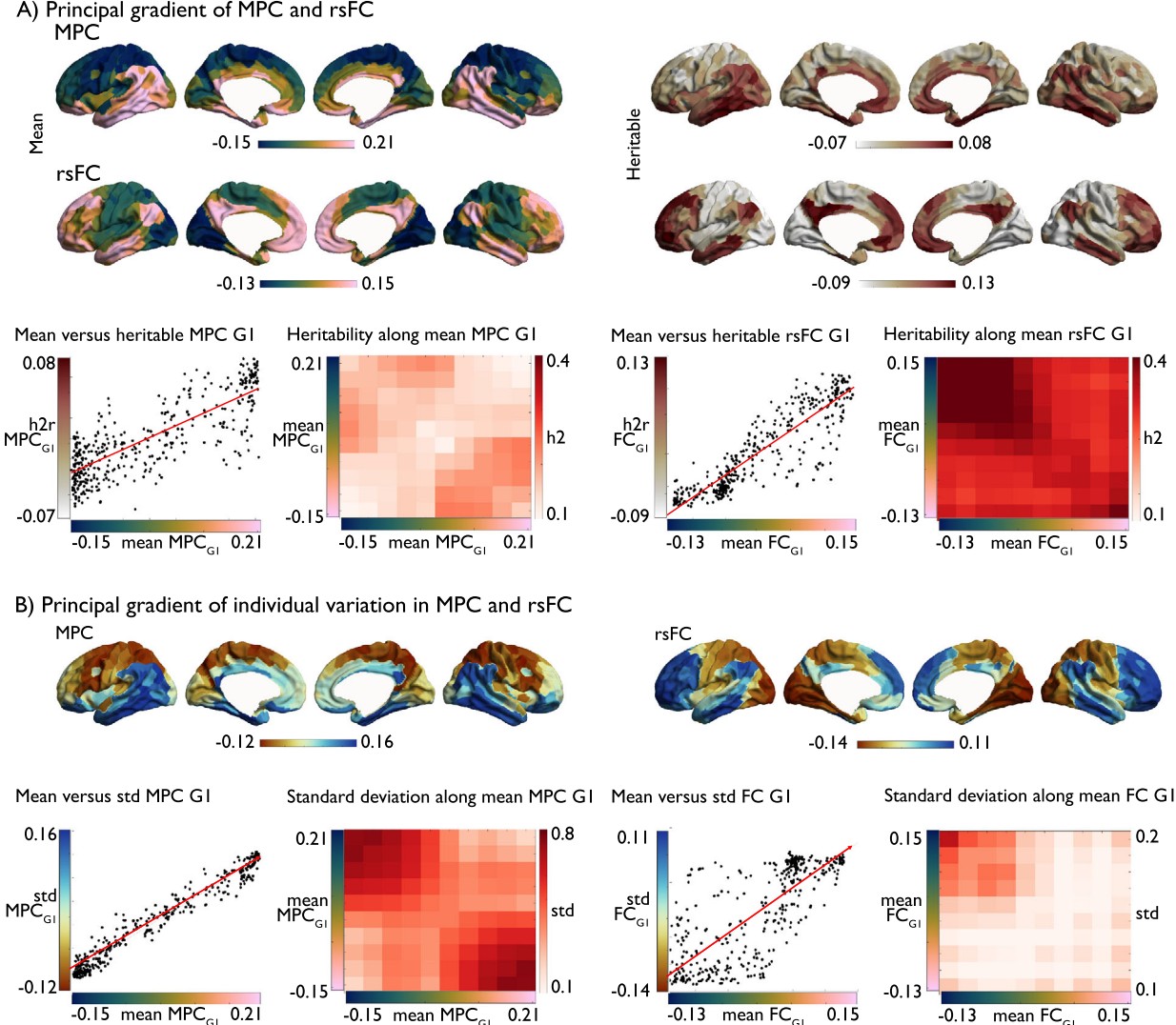

**Fig. 5 Principal gradient of heritability and individual variability. A** Principal gradient of MPC and rsFC, left: gradients based on mean data on MPC and rsFC and right gradients of heritable data alone, lower left panel: mean versus heritable MPC G1, as well as heritably along the principal mean gradient in MPC; lower right panel: mean versus heritable rsFC G1, as well as heritably along the principal mean gradient in rsFC; **B** Principal gradient of individual variation (std) in MPC and rsFC, lower panel left: correlation between mean and std MPC G1, and std along the mean gradient of MPC; lower panel right: correlation between mean and std rsFC G1, and std along the mean gradient of FC. Source data are provided as a Source Data file.

regions and related to terms such as "working memory", "attention", and "executive control" (left upper quadrant) and "action", as well as "multisensory processes" (right upper quadrant). Quadrants with low coupling included the paralimbic regions, which were related to affective processes ("emotion", "reward", "pain"; lower right), and, heteromodal regions, which were associated with abstract cognition ("episodic/autobiographical memory", "social-cognition", lower left). Moreover, we explored how individual differences in behavior, measured using task batteries included in HCP, were differentiated along the two axes identified in our study. In particular, decreased differentiation between principal gradients of MPC and rsFC related to "stress" and "pain", whereas increased difference related to "working memory" and "cognition". This observation suggests that the dimensions identified by our analysis may have behavioral relevance, an important hypothesis for future studies to examine (Supplementary Results).

## Discussion

Our study set out to understand how flexible cognition and behavior emerge from the interplay of cortical structure and function. The work was motivated by an emerging hypothesis that reductions in structure–function associations in transmodal regions of cortex may enable cognitive processes that are less constrained to specific modalities of information[1,11,21,22,24,25,29,65]. Our analysis confirmed prior observations that human brain structure and function are least coupled in transmodal cortex. We found a shared reduction in genetic control over structure and function of heteromodal cortex that included both default mode and fronto-parietal networks. Notably, this decoupling was not consistently present in paralimbic components of transmodal cortex, such as in anterior insula and cingulate regions. Analyses in macaques showed a similar structure–function decoupling, indicating that the overarching organization is likely broadly conserved across primate species[66]. However, the decoupling in

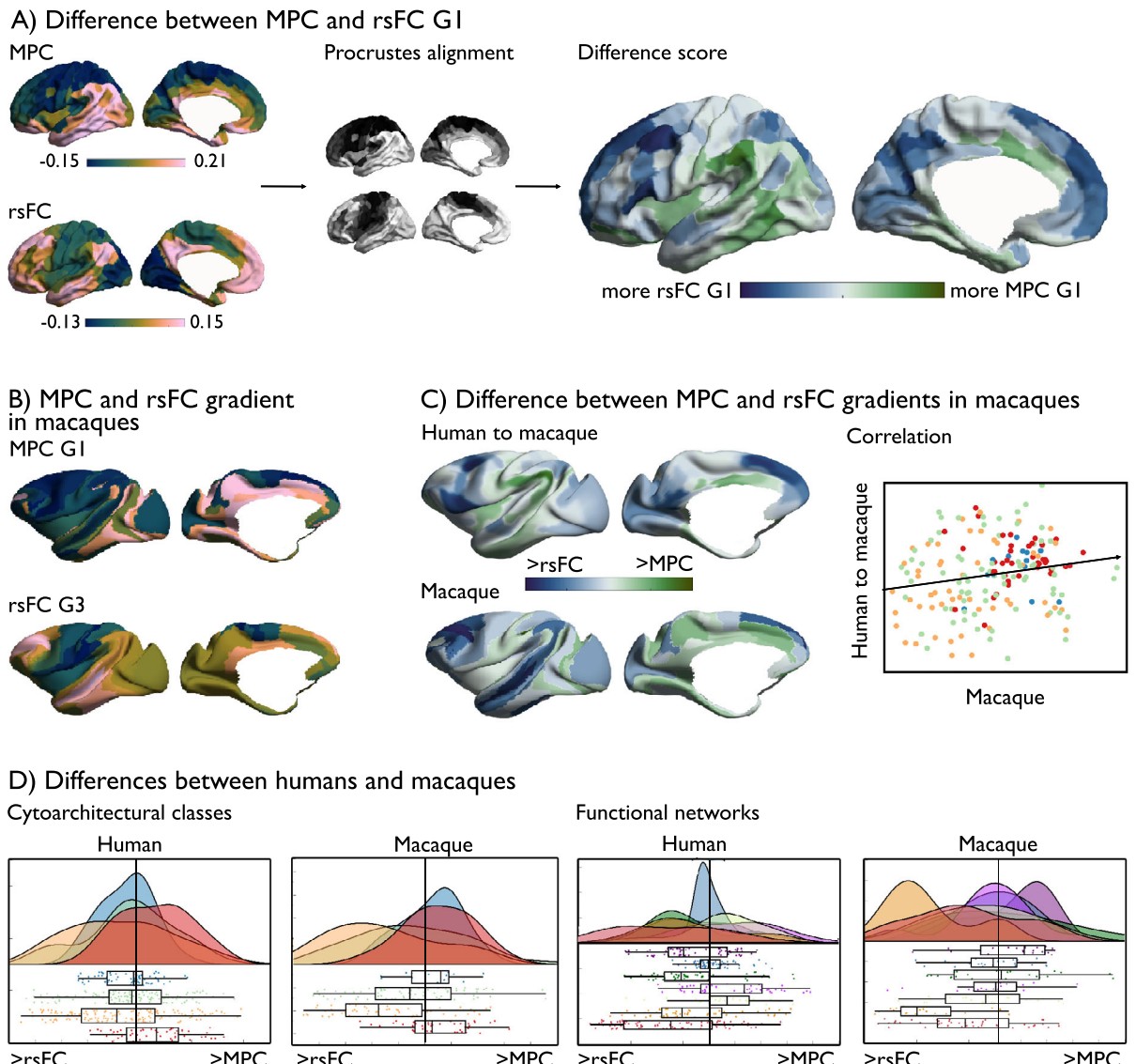

**Fig. 6 Difference in organizational gradients of MPC and rsFC in humans and macaques. A** Left panel: principal MPC and rsFC gradient; middle panel: alignment; right panel: $\Delta_{rsFC-MPCG1}$; **B** principal gradient of MPC and tertiary gradient of rsFC in macaques; **C** Difference between principal gradients of MPC and rsFC in humans, mapped to macaque space, and difference between corresponding gradients in macaques (lower panel); right: correlation between human and macaque maps; **D** Raincloud plots of organizational differences as a function of cytoarchitectural class[59], and functional networks[28] in humans (400 parcels, Schaefer parcellation) and macaques (182 parcels, Markov parcellation), boxes show the median and interquartile (25–75%) range, whiskers depict the 1.5*IQR from the quartile. Source data are provided as a Source Data file.

macaques was less pronounced than in humans, particularly in unimodal and heteromodal but not in paralimbic areas, an observation that warrants further investigation. Within transmodal cortex, our functional analysis indicated variable degrees of structure–function coupling: heteromodal regions were implicated in processes such as social-cognition and episodic memory, while paralimbic regions were more closely implicated in affective/motivational processes. Moreover, the topography of structure–function coupling related to different gene expression patterns, with regions with greater structure–function coupling associated with genes expressed later in development. Collectively, our assessment of how genetic and evolutionary factors contribute to cortical structure–function coupling disentangles in particular

different components of transmodal cortex, believed to be key to culturally enriched thought and, thus, human cognition.

The transmodal regions in which MPC and rsFC diverge are also recognized as those areas in where experience-induced development is heightened and plasticity is greatest[67]. Moreover, transitions from late childhood to early adulthood also show altered structure–function coupling in transmodal regions[27], and consistent changes in cortical microstructure and myelination patterns[9]. Protracted and reduced myelination of axons may aid the coordination of distributed neural activity later in life, allowing neural motifs to emerge over time that reflect the specific experiential constraints faced by the individual[27,68]. In the adult brain, regions of transmodal cortex are associated with higher

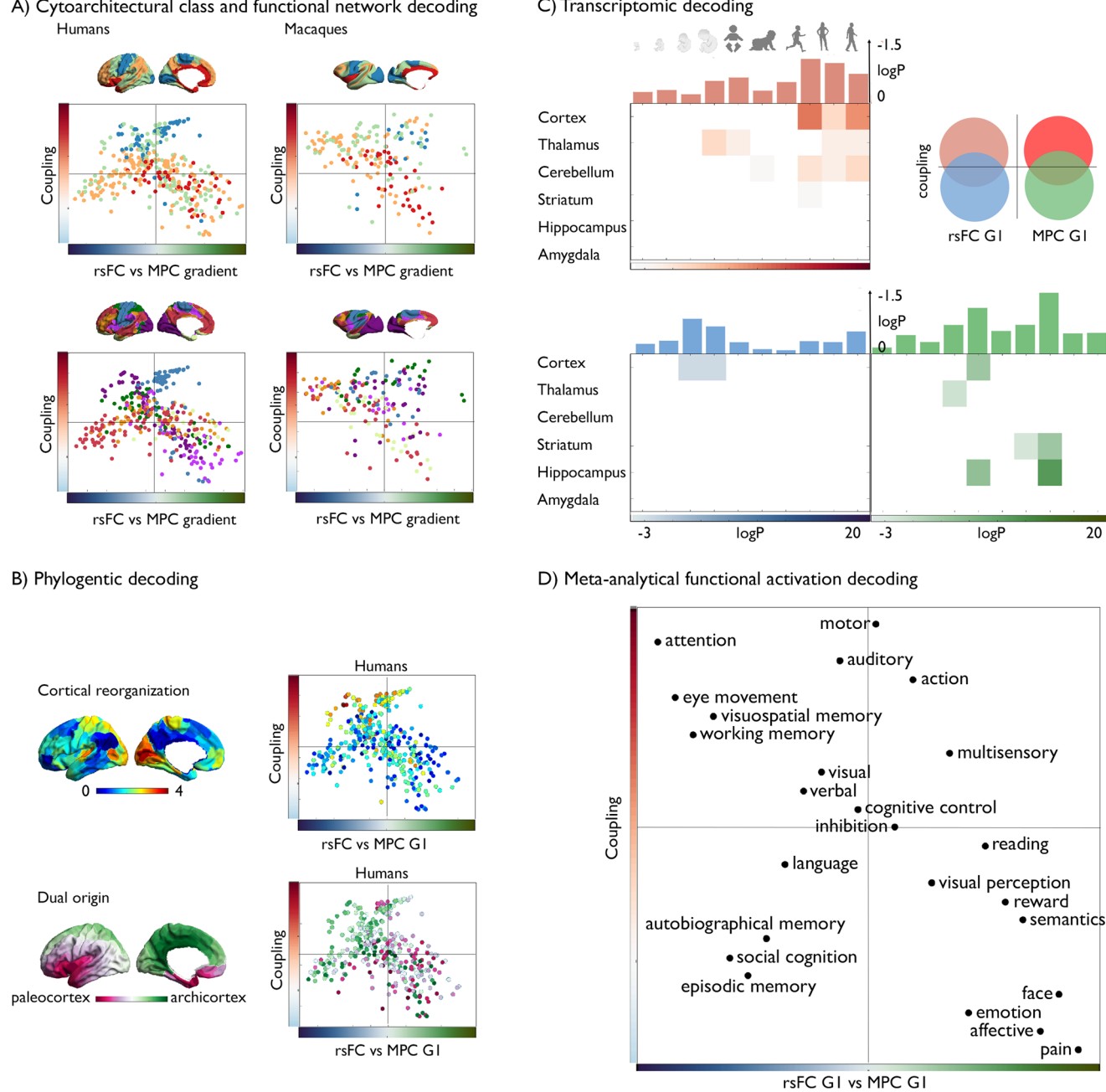

**Fig. 7 Multiscale quadrants of structure–function coupling. A** Cytoarchitectural class[59] and functional community[28] decoding along 2D model of the difference between microstructural and functional connectivity gradients in humans and macaques (x-axis) and microstructure-functional connectivity coupling (y-axes); **B** Phylogenetic models using cortical reorganization[41] and a model of dual patterning in the cerebral cortex[53]; **C** Transcriptomic developmental decoding of coupling and gradient differences between structure and function, Red/Blue/Green tones represent the log transformed false discovery rate (FDR)-corrected p-values [–20 –3]. The bar plot above represents the log transformed (FDR)-corrected p-values, averaged across all brain structures. Red indicates the genes that were attributed to the left upper quadrant, blue indicates the values were higher for the functional end of the difference gradient and untethered, whereas green reflects the microstructural apex and untethered. Only values that are below FDRq < 0.05 are displayed; **D** 2D projection of NeuroSynth meta-analysis[56] of regions of interest along $\Delta_{rsFC-MPCG1}$ map (x-axis) and structure–function uncoupling (y-axis) using 24 topic terms. We binned both $\Delta_{rsFC-MPCG1}$ and coupling maps in 20 equally sized bins, averaged the z-scores of meta-analytical activations per bin and ranked their weighted means along the x- and y-axes. Source data are provided as a Source Data file.

degrees of synapse formation and growth, as measured by aerobic glycolysis[69]. The apparent reduction in the genetic control over structure–function decoupling in heteromodal cortex established by our study may reflect the conditions that facilitate cortical adaptation to continue to occur after birth. Consistent with this view, regions within the default mode and fronto-parietal networks show most evidence of dynamic changes and experience-

dependent plasticity at short and longer time scales[70–72]. For example, studies of spontaneous cognition suggest that dynamic reconfigurations of default mode and fronto-parietal systems are linked to ongoing changes in patterns of thought[73,74]. At the same time, microcircuit models of cortical dynamics have been shown to benefit from integrating regionally varying myelin-sensitive MRI information, highlighting that cortical dynamics

vary as a function of local microstructural context[75]. This indicates a different coupling of microstructure and intrinsic function along the cortical hierarchy.

Our functional analysis defines potential constraints on human cognition that structure–function decoupling may support. Regions with maximal structure–function coupling were associated with cognitive terms such as "attention" and "working memory". Conversely, uncoupled regions that were similar in humans and macaques were linked to emotional and motivational states, processes that are important in most if not all mammals[1]. Notably, however, regions in which structure–function decoupling showed differences between human and macaques, and differences between structural and functional organization, were associated with functions such as "social-cognition" and "episodic/autobiographical memory". Although both declarative memory, and explicit social functions have been reported in non-human primates[76,77] these types of process are thought to be part of a set of processes that form the foundations of cultural learning in our species. Our study, therefore, adds to a growing body of work suggest that coordinated coupling of structure and function itself is under genetic control and that this likely determines variation in cognitive phenotypes[27,78]. Notably, studying heritability of coupling itself, we found heteromodal coupling to be least heritable, a pattern consistent with the spatial distribution of coupling heritability in previous work[78]. More generally, our analysis provides insights into the broader neural organization of human cognition. The microstructural gradient captures spatial shifts in cortical lamination patterns[22], representing a "sensory-fugal" axis running towards paralimbic cortices[1]. This axis has been implicated in predictive processing of allostatic needs and motivation[79]. Also anchored on sensory systems, but radiating towards heteromodal networks such as the default mode network, the principal functional gradient closely relates the distance of brain regions from sensory input[3,80,81] and may in part capture a divergent pattern of connectivity from that described by the sensory-fugal hierarchy. For example, heteromodal regions show the most long-range functional connectivity patterns, linked to changes in supragranular layers[82–84]. In this context, it is notable that heteromodal networks such as the default mode network are physically distant from unimodal systems, a topological principle that may allow for the integration of external and internal information[3,11]. Our findings are also consistent with the notion that rapid evolutionary expansion of the cerebral cortex may have shifted away from bottom-up activity cascades in primary sensory-motor regions towards a network of relatively untethered, long-distance, and increasingly parallelized heteromodal regions[11,41,84]. Interestingly, dissociations within transmodal systems are also aligned with distance from archi- and paleocortex. The dual origin theory of cortical organization[53,55] argues that neural differentiation progressively stems from an archicortical and paleocortical origin[53]. Based on our findings, it is possible that archicortical trends preferentially relate to heteromodal cortices, while paleocortical trends encompass paralimbic regions. Together, these observations suggest that differentiable genetic processes may underlie transmodal uncoupling in heteromodal and paralimbic cortices, an important hypothesis for future studies to examine.

We also observed different transcriptomic associations across the sensory-fugal axis more generally, and a divergence of heteromodal and paralimbic cortices within the transmodal system specifically[21,24,65]. Primary sensory regions with close coupling showed an association with postnatal expression of genes in neocortical regions, thalamus, and cerebellum, in line with cerebellar-thalamic-cortical circuits[85]. On the other hand, paralimbic transmodal regions were associated with prenatal genetic expression in the neocortical regions, thalamus, and hippocampus and with postnatal genetic expression in the striatum and hippocampus. Finally, heteromodal regions harbored genes expressed in neocortical regions in prenatal stages. The different expression time-windows observed may be a genetic indicator of how heteromodal regions, associated with genes expressed early in development, can develop richer connectivity profiles that are able to be transformed by experience, a pattern sometimes described as the "older get richer" principle[11,86]. Conversely, those regions associated with genetic expression later in development are under heightened genetic control, and not only include primary regions that show a tight coupling of structure and function—associated with cortical, thalamic, and cerebellar genetic expression, but also uncoupled paralimbic regions—associated with hippocampal and striatal genetic expression. Thus, transmodal decoupling may be understood from a perspective reaching beyond the cortex, guided by interactions with subcortical regions, such as the thalamus and striatum[85,87,88]. In particular, the thalamus alongside the cerebellar hemispheres, show recent evolutionary alterations[89,90] that are paralleled by increased connectivity to prefrontal cortices[91,92]. As such, it is possible that the uncoupling of particularly heteromodal regions observed in humans builds upon evolutionary alterations that shape not only in the cortex but also the subcortex, supporting key features of human cognition. It is of note that although the Allen human brain atlas is the most densely sampled transcriptomic dataset of the human brain to date, it comes with various limitations, such as sex and age imbalance[64,93]. Thus, findings from our study relating to gene expression should be interpreted with caution. In particular, although we selected only genes that were relatively consistent across donors, there may still be a bias toward those genes having higher expression rates in males than in females.

In the current study, we report genetic uncoupling in transmodal regions in humans using a twin-design and differences between humans and macaques in terms of structure–function associations. We were able to compare humans and macaques directly using previously established cross-species alignment techniques[41] in combination with histologically validated in vivo microstructure profile covariance analysis[22]. As it stands, meaningful comparisons between species using the current techniques are limited to these two species. Thus, although we could show that the patterning of structure–function uncoupling is consistent in humans and macaques, and therefore, likely important in many primates, we can only speculate about the evolutionary relevance of the observed differential uncoupling in unimodal and heteromodal regions across species. Anatomical differences in these regions, including frontal and parietal cortices, of other primates relative to humans have been reported, including differences in cortical and deep white matter architecture[15,40,94]. Though previous functional work has reported homologs of the default mode network in other primates beyond macaques[41], including marmosets[95] and chimpanzees[96], it has recently been reported that the functional network hierarchy is different in humans relative to macaques, with regions in the default mode regions showing most divergence[41]. One possible hypothesis is that there is an evolutionary adaptation in structure and function of unimodal and heteromodal association regions between humans and NHPs[41,95,97], resulting in progressive untethering of structure and function in the former that gives rise to aspects of cognition that are most developed in humans[11]. Secondly, structure–function coupling shows some association with individual variations such as sex and behavioral traits and states. Indeed, though uncoupling was similar in both sexes in humans and macaques, ours and previous work also report subtle differences in coupling between (human) males and females. Such patterns may reflect sex-biased brain development[98] and hormonal variations[99]. Moreover, previous work has indicated

differences in brain function at rest relate to the level of consciousness in humans and macaques[100,101], and also our findings suggest subtle differences in coupling of structure and function between awake and anesthetized macaques. Despite this variation, we observed a consistent difference in unimodal and heteromodal coupling between (awake) humans and awake as well as anesthetized macaques. Future work may expand on this work to further explore variations in structure–function coupling by level of consciousness, particularly in humans by exploiting the capacity for introspection. Such analysis may provide additional insight in the relation between structure–function coupling and features of internal cognition.

In conclusion, our study set out to understand how flexible cognition emerges from a genetically controlled microstructural architecture. Our analyses suggest that the previously documented pattern of reduced structure–function coupling in transmodal cortices is heritable in humans, and broadly conserved across primate species. In NHPs, this pattern was reduced relative to humans, and functional meta-analyses indicated that this difference was most likely associated with social-cognition and autobiographical memory. These data are consistent with the hypothesis that primate brain organization has evolved to support structure–function decoupling, which may facilitate types of cognition that benefit from learning and social interactions, enabling important features of cultural enrichment that are thought to be a characteristic of our species. To test this hypothesis, future work should expand our approach to include a wider range of species, and explore their association with different functional states in a more detailed manner. These studies will be an important step towards the understanding of how evolution has shaped neural function to support important features of human cognition, a research goal, which is likely to become increasingly feasible with growing availability of open multi-species in vivo datasets[38,39,95].

## Methods
The current research complies with all relevant ethical regulations as set by The Independent Research Ethics Committee at the Medical Faculty of the Heinrich-Heine-University of Duesseldorf (study number 2018–317).

### HCP sample
*Participants and study design.* We used publicly available data from the Human Connectome Project (HCP) S1200 release (http://www.humanconnectome.org/), which comprised data from 1206 individuals (656 females) that are made up by 298 MZ twins, 188 DZ twins, and 720 singletons, with age mean ± SD = 28.8 ± 3.7 years (age range = 22–37 years). The informed consent for all subjects was obtained by HCP. Our data usage was approved by HCP, and complies with all relevant ethical regulations for work with human participants. We included individuals for whom the scans and data had been released after passing the HCP quality control and assurance standards. The full set of inclusion and exclusion criteria are described elsewhere[7,50]. In short, the primary participant pool comes from healthy individuals born in Missouri to families that include twins, based on data from the Missouri Department of Health and Senior Services Bureau of Vital Records. Additional recruiting efforts were used to ensure participants broadly reflect ethnic and racial composition of the U.S. population. Healthy is broadly defined, in order to gain a sample generally representative of the population at large. Sibships with individuals having severe neurodevelopmental disorders (e.g., autism), documented neuropsychiatric disorders (e.g., schizophrenia or depression) or neurologic disorders (e.g., Parkinson's disease) are excluded, as well as individuals with diabetes or hypertension. Twins born prior to 34 weeks of gestation and non-twins born prior to 37 weeks of gestation were excluded. After removing individuals with missing structural and functional imaging data, our sample consisted of 992 (529 females) individuals (including 255 MZ twins and 150 DZ twins) with an age mean ± SD = 28.71 ± 3.72 years (range = 22–37 years).

*Structural-imaging processing.* MRI protocols of the HCP are previously described[7,50]. In short, MRI data used in the study were acquired on the HCP's custom 3T Siemens Skyra equipped with a 32-channel head coil. Two T1w images with identical parameters were acquired using a 3D-MP-RAGE sequence (0.7 mm isovoxels, matrix = 320 × 320, 256 sagittal slices; TR = 2400 ms, TE = 2.14 ms, TI = 1000 ms, flip angle = 8°; iPAT = 2). Two T2w images were acquired using a 3D T2-SPACE sequence with identical geometry (TR = 3200 ms, TE = 565 ms,

variable flip angle; iPAT = 2). T1w and T2w scans were acquired on the same day. The pipeline used to obtain the Freesurfer-segmentation is described in detail in a previous article[7] and is recommended for the HCP data. The preprocessing steps included co-registration of T1- and T2-weighted scans, B1 (bias field) correction, and segmentation and surface reconstruction using FreeSurfer version 5.3-HCP. Using these data, the equidistant surfaces are computed for MPC measurement.

*Parcellation approach.* For our main analysis, we used the Schäfer parcellation scheme[58] that combines local gradient and global similarity approaches via gradient-weighted Markov Random models. The parcellation has been evaluated with regards to stability and convergence with histological mapping and alternative parcellations. We focused on a 400-parcel atlas, but additionally evaluated results using the Glasser 360 atlas[44].

*Cortical microstructure and microstructural covariance networks.* We estimated MPC using myelin-sensitive MRI, in line with the previously reported protocol[22]. The myelin-sensitive contrast was T1w/T2w from the HCP minimal processing pipeline, which uses the T2w to correct for inhomogeneities in the T1w image. The MPC approach complements mean T1w/T2w ratio mapping, tapping into microstructural profiles across cortical depths. Depth-dependent cortical microstructure analysis has a long tradition in neuroanatomy[102,103], and depth-dependent shift in cellular and myelin characteristics have been shown to reflect architectural complexity[103] and cortical hierarchy[1]. We generated 12 equivolumetric surfaces between outer and inner cortical surfaces. The equivolumetric model compensates for cortical folding by varying the Euclidean distance $\rho$ between pairs of intracortical surfaces throughout the cortex to preserve the fractional volume between surfaces. $\rho$ was calculated as follows for each surface (1):

$$\rho = \frac{1}{A_{\text{out}} - A_{\text{in}}} \cdot \left( -A_{\text{in}} + \sqrt{\alpha A_{\text{out}}^2 + (1-\alpha)A_{\text{in}}^2} \right) \quad (1)$$

in which $\alpha$ represents a fraction of the total volume of the segment accounted for by the surface, while $A_{\text{out}}$ and $A_{\text{in}}$ represents the surface area of outer and inner cortical surfaces, respectively. We systematically sampled T1w/T2w values for each of the 64,984 linked vertices from the outer to the inner surface across the whole cortex. Subsequently, we computed the average value of T1w/T2 in each of the 400 parcels[58]. In turn, MPC (i,j) for a given pair of parcels i and j is defined by (2):

$$\text{MPC}(i,j) = \frac{1}{n} \sum_{s=1}^{n} \left( \frac{r_{ij} - r_{ic}r_{jc}}{\sqrt{\left(1 - r_{ic}^2\right)\left(1 - r_{jc}^2\right)}} \right)_s \quad (2)$$

in which $s$ is a participant and $n$ is the number of participants, $r_{ic}$ the correlation coefficient of the intensity profile at node $i$ with the average intensity profile across the entire cortex, and $r_{jc}$ the correlation of the intensity profile at node $j$ with the average intensity profile across the cortex. We used the MPC for further analysis.

*Functional connectivity.* Functional connectivity matrices were based on 1-h of resting-state fMRI data acquired through the HCP[7], which underwent HCP's minimal preprocessing[7,44]. Briefly, for each individual, a functional connectivity matrix was calculated using the correlation coefficient of the average of the four minimally preprocessed, spatially normalized, and concatenated to 4 15-min resting-state fMRI scans which were co-registered using MSMAll to template HCP 32k_LR surface space[50]. 32k_LR surface space consists of 32,492 total nodes per hemisphere (59,412 excluding the medial wall). Following average timeseries were extracted in each of the 400 cortical parcels[58] and individual functional connectivity matrices were computed. The individual functional connectomes were generated by averaging preprocessed timeseries within nodes, correlating nodal timeseries and converting them to z scores. Here we used the individual timeseries of individuals with complete data in the S1200 sample.

*Measuring structure–function coupling.* Structure–function coupling was defined as the row-wise correlation between the mean MPC and rsFC connectomes. In short, the values of each row in two matrices were selected and correlated; next, the Spearman's rho-value resulting from this analysis was projected on the surface. A similar approach quantified genetic coupling, i.e., we correlated row-wise mean MPC/rsFC with the row-wise heritability maps.

*Heritability analysis.* To investigate heritability of MPC and intrinsic functional connectomes, we analyzed edge-wise connectomes of both measures in a twin-based analysis. The quantitative genetic assessments were implemented with Sequential Oligogenic Linkage Analysis Routines (SOLAR)[104]. This toolbox uses maximum likelihood variance-decomposition methods to assess the relative importance of genetic-vs-environmental (G + E) influences on a phenotype by modeling the covariance among family members as a function of genetic proximity. Based on previous work indicating G + E is more parsimonious and leads to more reproducible results in HCP, we used a G + E model to assess heritability in this sample[105]. SOLAR can handle pedigrees of arbitrary size and complexity. To ensure that our functional connectivity and microstructural profile covariance

measures were conform to the assumptions of normality, an inverse normal transformation was applied, similar to previous work[106].

Heritability ($h^2$) reflects the share of the phenotypic variance ($\sigma^2_p$) accounted for by the total additive genetic variance ($\sigma^2_g$), i.e., $h^2 = \sigma^2_g/\sigma^2_p$. Phenotypes showing stronger covariances between genetically more similar individuals than between genetically less similar individuals have higher heritability. SOLAR contrasts the detected covariance matrices, here node-wide rsFC/MPC and coupling of rsFC and MPC, with the structure of the covariance matrix predicted by kinship. We studied heritability with simultaneous estimation for the effects of potential covariates. For this study, we included covariates including age, sex, age[2], and age * sex. To evaluate robustness of our heritability findings of rsFC with respect to intra-individual variation we additionally computed heritability considering possible within-subject variation described in previous work[61,78].

*Macaque data.* All datasets in this study were from openly available sources. The macaque data used for our main analyses stemmed from one cohort (University of California, Davis of the recently established PRIME-DE (https://fcon_1000.projects.nitrc.org/indi/indiPRIME.html)[38]. The full dataset consisted of 19 rhesus macaque monkeys (macaca mulatta, all female, age ± SD = 20.38 ± 0.93 years, weight = 9.70 ± 1.58 kg) scanned on a Siemens Skyra 3T with 4-channel clamshell coil. All the animals were scanned under anesthesia. In brief, the macaques were sedated with injection of ketamine (10 mg/kg), dexmedetomidine (0.01 mg/kg), and buprenorphine (0.01 mg/kg). The anesthesia was maintained with isoflurane at 1–2%. The details of the scan and anesthesia protocol can be found at (https://fcon_1000.projects.nitrc.org/indi/PRIME/ucdavis.html). The neuroimaging experiments and associated procedures were performed at the California National Primate Research Center (CNPRC) under protocols approved by the University of California, Davis Institutional Animal Care and Use Committee[107]. The resting-state fMRI data were collected with 1.4 × 1.4 × 1.4 mm resolution, TR = 1.6 s, 6.67 min (250 volumes) under anesthesia. No contrast-agent was used during the scans. Structural data (T1w and T2w) were acquired with 0.3 × 0.6 × 0.6 mm resolution, with interpolation on to generate 0.3 mm isotropic resolution (T1w: TR = 2500 ms, TE = 3.65 ms, TI = 1100 ms, flip angle = 7 degrees, FOV = 154 mm; T2w: TR = 3000 ms, TE = 307 ms).

To evaluate robustness of the macaque findings, we evaluated two additional macaque samples with rsFC data openly available in PRIME-DE (https://fcon_1000.projects.nitrc.org/indi/indiPRIME.html)[38], previously used in ref. [41]. The samples varied in age-range, sex, and acquisition parameters. The Newcastle dataset (awake[108,109]) consisted of 10 rhesus macaques (8 males, age mean ± SD = 8.28 ± 2.33, weight = 11.76 ± 3.38) scanned on a Vertical Bruker 4.7 T primate scanner. The fMRI session was acquired with 1.2 × 1.2 × 1.2 mm resolution, TR = 2000 ms, 8.33-min per scan (250 volumes × 2 scan) per animal. No contrast-agent was used during the scans. In the case of the Oxford dataset (anesthetized), we used nineteen rhesus macaques with preprocessing and surface reconstruction as in previous work[41] (all males, age = 4.01 ± 0.98 years, weight = 6.61 ± 2.04 kg). The macaques were scanned on a 3T with a 4-channel coil[110]. Resting-state fMRI (rs-fMRI) data were collected with 2 mm isotropic resolution, TR = 2000 ms, 53.3 min (1600 volumes). No contrast-agent was used during the scans.

*MRI data processing.* The structural T1w and T2w images were preprocessed using the customized HCP-like macaque pipeline (doi:10.5281/record/zenodo.3888969). In short, the preprocessing includes (1) spatial denoising by a non-local mean filtering operation[111], (2) brain extraction using ANTs registration with a reference brain mask followed by manually editing to fix the incorrect volume (ITK-SNAP, https://www.itksnap.org)[112]; (3) tissue segmentation using ANTs joint label fusion algorithm and surface reconstruction (FreeSurfer)[113]; (4) T1w and T2w alignment (linear) followed by the linear and non-linear registration to the high-resolution template space (0.3 mm); (5) the native white matter and pial surfaces were registered to the Yerkes19 macaque surface template[114,115]. MPC was computed similarly to the human approach, however due to the thinner cortex in macaques relative to humans we decided to only focus on 9 equidistant surfaces. The macaque monkey intrinsic functional data were preprocessed using a customized Connectome Computational System pipeline for nonhuman primates[41]. Briefly, the rs-fMRI data were preprocessed including temporal de-spiking, motion correction, 4D global scaling, nuisance regression using white matter (WM), and cerebrospinal fluid (CSF) signal and Friston-24 parameter models, bandpass filtering (0.01–0.1 Hz), detrending and co-registration to the native anatomical space. The data were then projected to the native mid-cortical surface and smoothed along the surface with FHWM=3 mm. Finally, the preprocessed data were down-sampled to a standard 10k (10,242 vertices) resolution surface[114]. Similar with human pre-processing, functional timeseries were averaged within the Markov parcellation[116], and a connectivity matrix was constructed.

*Alignment of human to macaque space.* To evaluate the similarity between human and macaque cortical patterns we transformed the human pattern (rsFC-MPC coupling/gradients) to macaque cortex based on a functional-alignment techniques recently developed. This method leverages advances in representing functional organization in high-dimensional common space and provides a transformation between human and macaque cortices[41]. This, enabled us to directly compare between species within the same space.

*Gradient decomposition.* To compute macroscale gradients, we performed several analysis steps. The input of the analysis was the MPC and rsFC matrix, cutoff at 90% similar to previous studies[3,22]. To study the relationships between cortical regions in terms of their features, we used a normalized angle similarity kernel resulting in a non-negative square symmetric affinity matrix. Next, we used diffusion mapping, a non-linear dimensionality reduction method[62]. The algorithm estimates a low-dimensional embedding from a high-dimensional affinity matrix. In this space, cortical nodes that are strongly interconnected by many supra-threshold edges or few very strong edges are closer together, whereas nodes with little or no covariance are farther apart. The name of this approach, which belongs to the family of graph Laplacians, derives from the equivalence of the Euclidean distance between points in the embedded space and the diffusion distance between probability distributions centered at those points. It is controlled by a two parameters α and t, where α controls the influence of the density of sampling points on the manifold (α = 0, maximal influence; α = 1, no influence), and t controls the scale of eigenvalues. Based on previous work[3,22] we followed recommendations and set α = 0.5 and t = 0, a choice that retains the global relations between data points in the embedded space and has been suggested to be relatively robust to noise. Gradient parameters were identical for human and macaque data. To evaluate robustness of the gradient measures we varied alpha between 0 and 1 with steps of 0.1 and diffusion time between 0 and 9 with steps of 1 in humans and macaque rsFC and MPC data.

*Functional decoding.* For functional decoding, we selected 24 behavioral paradigms as previously reported[3,22], averaged in the 400 Schaefer parcels. To perform functional decoding, we averaged the z-scores of NeuroSynth topic terms along the pattern of interest across 20 equally sizes bins and performed weighted averaging, and ranked the values accordingly to capture which functions are associated with MPC-rsFC uncoupling and the difference between MPC and rsFC gradients. Following the ranking of the tasks is projected in 2D space.

To assess the association between structure–function coupling and gradient difference and individual-level behavior, we selected 20 markers of individual difference from the HCP battery i.e., total cognition, card sorting, list sorting, friendship, picture vocabulary, reading English, pain, endurance, flanker, picture sequence, self-efficiency, perceived stress, noise, SCPT, sadness, and NEO-FFI[50]. Subsequently, we performed linear regression using SurfStat[117] to probe association between parcel-level variation in coupling and gradient difference, while controlling for age and sex. Then we correlated the 20-task-based t-maps with mean maps of coupling difference and gradient differences and visualized them in 2D space to evaluate their relationship to patterns of coupling and gradient difference.

*Comparisons between gradients and modalities.* To assess correlations between spatial maps, we used spin-tests to control for spatial autocorrelation when possible[118].

*Phylogenetic maps of cortical reorganization and archi-paleocortex distance.* To perform phylogenetic decoding, we used cortical reorganization between macaque monkeys and humans[41] (https://github.com/tingsterx/alignment_macaque-human), as well as a model of the dual origin, similar to previous work[106]. Here, we combined the distance to paleo- and archicortex in one map, assigning each parcel with the distance closest to either origin.

*Transcriptomic association analysis.* Given the association between phylogeny and ontogeny[21], we correlated both maps with post-mortem gene expression data from the Allen Human Brain Atlas (AHBA)[64] and evaluated the spatiotemporal time windows in which these genes are most frequently expressed using developmental gene set enrichment analysis[119]. We assessed spatial correlations of structure–function coupling and the difference between large-scale principal gradients of MPC and rsFC and gene expression patterns. First, we correlated the t-statistics map of the two axes with the post-mortem gene expression maps provided by Allen Institute for Brain Sciences (AIBS) using Neurovault gene decoding[64,120]. Neurovault implements mixed-effect analysis to estimate associations between the input map and the genes of AIBS donor brains yielding the gene symbols associated with the input map. Gene symbols that passed a significance level of FDR-corrected $p < 0.05$ were further tested whether they are consistently expressed across the donors using abagen (https://github.com/rmarkello/abagen), which implements prior recommendations for imaging-transcriptomics studies[93]. For each gene, we estimated the whole-brain expression map and correlated it between all pair of different donors. Only genes showing consistent whole-brain expression pattern across donors ($r > 0.5$) were retained. In a second stage, gene lists that were significant were fed into enrichment analysis, which involved comparison against developmental expression profiles from the BrainSpan dataset (http://www.brainspan.org) using the cell-type-specific expression analysis (CSEA) developmental expression analysis tool (http://genetics.wustl.edu/jdlab/csea- tool-2)[119]. As the AIBS repository is composed of adult post-mortem datasets, it should be noted that the associated gene symbols represent indirect associations with the developmental data.

## Replication and robustness

*MICs dataset.* Participants: Data were collected in a sample of 50 healthy volunteers (21 women; age mean ± SD = 29.82 ± 5.73 years; 47 right-handed) between April 2018

and September 2020[60]. Each participant underwent a single testing session. All participants denied a history of neurological and psychiatric illness. The Ethics Committee of the Montreal Neurological Institute and Hospital approved the study. Written informed consent, including a statement for openly sharing all data in anonymized form, was obtained from all participants.

*MRI data acquisition.* Scans were completed at the Brain Imaging Centre of the Montreal Neurological Institute and Hospital on a 3T Siemens Magnetom Prisma-Fit equipped with a 64-channel head coil. Participants underwent a T1-weighted (T1w) structural scan, followed by resting-state functional MRI (rs-fMRI). In addition, a pair of spin-echo images was acquired for distortion correction of individual rs-fMRI scans. A second T1w scan was then acquired, followed by qT1 mapping. Total scan time for these acquisitions was ~45 min.

Two T1w scans with identical parameters were acquired with a 3D magnetization-prepared rapid gradient-echo sequence (MP-RAGE; 0.8 mm isovoxels, matrix = 320 × 320, 224 sagittal slices, TR = 2300 ms, TE = 3.14 ms, TI = 900 ms, flip angle = 9°, iPAT = 2, partial Fourier = 6/8). Both T1w scans were visually inspected to ensure minimal head motion before they were submitted to further processing. qT1 relaxometry data were acquired using a 3D-MP2RAGE sequence (0.8 mm isovoxels, 240 sagittal slices, TR = 5000 ms, TE = 2.9 ms, TI 1 = 940 ms, T1 2 = 2830 ms, flip angle 1 = 4°, flip angle 2 = 5°, iPAT = 3, bandwidth = 270 Hz/px, echo spacing = 7.2 ms, partial Fourier = 6/8). We combined two inversion images for qT1 mapping in order to minimize sensitivity to B1 inhomogeneities and optimize intra- and intersubject reliability[121,122]. One 7 min rs-fMRI scan was acquired using multiband accelerated 2D-BOLD echo-planar imaging (3 mm isotropic voxels, TR = 600 ms, TE = 30 ms, flip angle = 52°, FOV = 240 × 240mm², slice thickness = 3 mm, mb factor = 6, echo spacing = 0.54 ms). Participants were instructed to keep their eyes open, look at a fixation cross, and not fall asleep. We also include two spin-echo images with reverse phase encoding for distortion correction of the rs-fMRI scans (3 mm isotropic voxels, TR = 4029 ms, TE = 48 ms, flip angle = 90°, FOV = 240 × 240mm², slice thickness = 3 mm, echo spacing = 0.54 ms, phase encoding = AP/PA, bandwidth = 2084 Hz/Px).

*MRI data preprocessing.* Raw DICOMS were sorted by sequence into distinct directories using custom scripts. Sorted files were converted to NIfTI format using dcm2niix (v1.0.20200427; https://github.com/rordenlab/dcm2niix)[123], renamed, and assigned to their respective subject-specific directories according to BIDS standards[124]. Agreement between the resulting data structure and BIDS standards was ascertained using the BIDS-validator (v1.5.10; DOI: 10.5281/zenodo.3762221). All further processing was performed via micapipe, an openly accessible processing pipeline for multimodal MRI data (https://micapipe.readthedocs.io/). As previously described[60], rs-fMRI data were preprocessed using a combination of AFNI[125] and FSL[126]. To ensure magnetic field saturation, the first five volumes were disregarded. Images were reoriented, motion and distortion corrected. Motion correction was performed by registering all timepoints to the mean volume, while distortion correction leveraged main phase and reverse phase field maps acquired alongside rs-fMRI scans. Nuisance variable signal was removed using an in-house trained ICA-FIX classifier[127] and via spike regression using motion outlier outputs provided by FSL. Volumetric timeseries were averaged for registration to native FreeSurfer space using boundary-based registration[128], and mapped to individual surface models using trilinear interpolation. Native surface cortical timeseries underwent spatial smoothing once mapped to each individual's cortical surface models (Gaussian kernel, FWHM=10 mm)[7,129], and were subsequently averaged within nodes defined by the Schaefer 400 parcellation.

*eNKI dataset.* To evaluate whether differences in rs-fMRI preprocessing between humans and macaques can explain differences observed in coupling of MPC and rsFC between both species, we evaluated structure–function coupling in an additional human sample (eNKI subsample. N = 100, age-range 18–40 years) preprocessed analogously to the macaque datasets, including temporal compression, motion correction, 4D global scaling, nuisance regression (Friston's 24 model, cerebrospinal fluid and white matter), linear and quadratic detrends, bandpass filtering (0.01–0.1 Hz), and surface registration (for details, see ref.[130]). The NKI dataset was obtained from the publicly shared enhanced Nathan Kline Institute-Rockland Sample data repository.

*Test–retest reliability.* To evaluate test–retest reliability of our measures we used the test–retest sample of HCP[7] with complete T1wT2w and rsFC data at baseline and retest, leaving us with n = 46, 32 females, age: mean/SD 30.196/3.364, range: 22–35, mean days of test–retest interval: 139.304 days, SD 68.994, min-max: 18–343 days. ICC was quantified using a Median Absolute Deviation Intraclass Correlation Coefficient; https://warwick.ac.uk/fac/sci/statistics/staff/academic-research/nichols/scripts/matlab/madicc.m.

**Reporting summary**. Further information on research design is available in the Nature Research Reporting Summary linked to this article.

## Data availability

This study followed institutional review board guidelines of corresponding institutions. Human data analyzed in our main results were obtained from the open-access HCP S1200 young adult sample (HCP; http://www.humanconnectome.org/). MICs replication data is openly available at https://portal.conp.ca/dataset?id=projects/mica-mics. Macaque data was obtained from PRIME-DE (https://fcon_1000.projects.nitrc.org/indi/indiPRIME.html; University of California, Davis). Heritability analyses were performed using Solar Eclipse 8.4.0 (https://www.solar-eclipse-genetics.org), and data on the pedigree analysis is available here: https://www.nitrc.org/projects/se_linux/[104,131]. Gradient mapping analyses was based on BrainSpace (https://brainspace.readthedocs.io/en/latest/). Transcriptomic association analyses were conducted using NeuroVault (https://neurovault.org), abagen tools (https://github.com/rmarkello/abagen)[93], and cell-type-specific expression analysis (CSEA) (http://genetics.wustl.edu/jdlab/csea-tool-2)[119] Supplementary analysis were performed using the MICS dataset (https://portal.conp.ca/dataset?id=projects/mica-mics) and eNKI dataset (https://fcon_1000.projects.nitrc.org/indi/enhanced/). Source data are provided with this paper and code for visualization of parcel results on surface linked in our study's Github repository (https://github.com/CNG-LAB/structure_function). Source data are provided with this paper.

## Code availability

Code for generation of MPC is available at (https://github.com/MICA-MNI/MPC, now https://github.com/MICA-MNI/micapipe). Heritability analyses were performed using Solar Eclipse 8.4.0 (http://www.solar-eclipse-genetics.org), and data on the pedigree analysis is available here: https://www.nitrc.org/projects/se_linux/. The code for connectome gradient generation are available at https://github.com/MICA-MNI/BrainSpace. Transcriptomic association analyses were conducted using NeuroVault (https://neurovault.org), cell-type-specific expression analysis (CSEA) (http://genetics.wustl.edu/jdlab/csea-tool-2), and abagen tools (https://github.com/rmarkello/abagen). Further code to visualize the data is available at (https://github.com/CNG-LAB/structure_function) and Zenodo: (https://doi.org/10.5281/zenodo.6406172).

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

## Acknowledgements

We would like to thank the various contributors to the open-access databases that our data was downloaded from. HCP data were provided by the Human Connectome Project, Washington University, the University of Minnesota, and Oxford University Consortium (Principal Investigators: David Van Essen and Kamil Ugurbil;1U54MH091657) funded by the 16 NIH Institutes and Centers that support the NIH Blueprint for Neuroscience Research; and by the McDonnell Center for Systems Neuroscience at Washington University. For enhanced NKI, we would like to thank the principal support for the enhanced NKI-RS project is provided by the NIMH BRAINS R01MH094639-01 (PI Milham). Funding for key personnel was also provided in part by the New York State Office of Mental Health and Research Foundation for Mental Hygiene. Funding for the decompression and augmentation of administrative and phenotypic protocols provided by a grant from the Child Mind Institute (1FDN2012-1). Additional personnel support provided by the Center for the Developing Brain at the Child Mind Institute, as well as NIMH R01MH081218, R01MH083246, and R21MH084126. Project support also provided by the NKI Center for Advanced Brain Imaging (CABI), the Brain Research Foundation (Chicago, IL), and the Stavros Niarchos Foundation. This study was supported by the Deutsche Forschungsgemeinschaft (DFG, EI 816/21-1), the National Institute of Mental Health (R01-MH074457), the Helmholtz Portfolio Theme "Supercomputing and Modeling for the Human Brain" and the European Union's Horizon 2020 Research and Innovation Program under Grant Agreement No. 785907 (HBP SGA2). S.L.V. was supported by Max Planck Gesellschaft (Otto Hahn award). B.P. was funded by the National Research Foundation of Korea (NRF2021R1F1A1052303), Institute for Information and Communications Technology Planning and Evaluation (IITP) funded by the Korea Government (MSIT) (2020-0-01389, Artificial Intelligence Convergence Research Center, Inha University; 2021-0-02068, Artificial Intelligence Innovation Hub), and Institute for Basic Science (IBS-R015-D1). B.T.T.Y. is supported by the Singapore National Research Foundation (NRF) Fellowship (Class of 2017), the NUS Yong Loo Lin School of Medicine (NUHSRO/2020/124/TMR/LOA) and the Singapore National Medical Research Council (NMRC) LCG (OFLCG19-May-0035). Any opinions, findings, and conclusions or recommendations expressed in this

material are those of the authors and do not reflect the views of the Singapore NRF or the Singapore NMRC. B.C.B. acknowledges support from the SickKids Foundation (NI17-039), the National Sciences and Engineering Research Council of Canada (NSERC; Discovery-1304413), CIHR (FDN-154298), Azrieli Center for Autism Research (ACAR), an MNI-Cambridge collaboration grant, and the Canada Research Chairs program. C.P. was funded through a postdoctoral fellowship of the Fonds de la Recherche due Quebec–Santé (FRQ-S). Last, this work was funded in part by Helmholtz Association's Initiative and Networking Fund under the Helmholtz International Lab grant agreement InterLabs-0015, and the Canada First Research Excellence Fund (CFREF Competition 2, 2015–2016) awarded to the Healthy Brains, Healthy Lives initiative at McGill University, through the Helmholtz International BigBrain Analytics and Learning Laboratory (HIBALL), including S.L.V., C.P., S.B.E., and B.C.B.

## Author contributions

S.L.V., S.B.E., and B.C.B. conceptualized the work; S.L.V., S.B.E., B.C.B. gave input on analysis; S.L.V. performed the main analysis; T.X. performed microstructural profile analysis on the macaque data and computed macaque-human gradient correspondence; C.P. and B.C.B. provided the methods of the microstructural profile covariance; J.R. processed and provided the mics replication data; P.K. contributed to genetic correlation analysis of the twin model; S.L.V.; J.S.; S.B.E.; and B.C.B. drafted the manuscript; T.X., C.P., B.P., R.B., R.W., J.R., S.K.M., S.S.B., P.K., B.T.T.Y., D.M. contributed to revising the manuscript.

## Funding

## Competing interests

The authors declare no competing interests.
