## [Peer Review File · Nature Communications]

Genetic and phylogenetic uncoupling of structure and function in human transmodal cortexREVIEWER COMMENTS

Reviewer #1 (Remarks to the Author):

This study combines heritability and cross-species (human and macaques) analyses to assess how changes in function-structure brain associations underpin cognition. The paper presents confirmatory findings (e.g., reduced function-structure coupling from sensory, unimodal, and transmodal regions) and some novel results (e.g., genetic control varies across the function-structure association axis; patterns of function-structure associations overlap in both species, with macaques showing a reduced decoupling in heteromodal but not paralimbic brain regions). Overall, the paper is well written, and its structure easy to follow, with figures that are clear and informative, and the methods state-of-the art. However, at this stage, I have the following concerns that seem to preclude an unequivocal interpretation of the key results:

1. The increased function-structure association in macaques relative to humans is an interesting finding. However, I have the following worries:
 - a. Could this result be explained by differences in the quality of data between the species (e.g., better signal to noise ratio in macaque data compared to humans)?
 - b. Could the suppressed brain activity (via anesthesia) in macaques explain the result (e.g., increased mapping of function on structure due to reduced variability)?
 - c. Are human and macaque data age-equivalent? If not, how are different neurodevelopmental states accounted for?
2. Expanding on my previous comment: The pre-processing of human and macaques resting-state data appears quite different- e.g., ICA FIX in humans, temporal filter and smoothing in macaques. The fact that a number of results match across species despite these differences can be seen as a strength, but such differences make the interpretation of across-species differences problematic. Replicating the analyses with different data from a different primate species (e.g., marmoset; Liu C. et al. Nature Comm., 2019, perhaps adopting a proxy measure of T1w/T2w if not available) could help to address this concern. The authors may come up with other confirmatory approaches to address the concern.
3. Are the findings robust against differences in individual-specific data and brain volume (e.g., coupling-brain volume)?
4. Gradient decomposition: How robust are the results to changes in the parameters alpha and t? Were these parameters identical for the analysis of human and macaque connectomes? If not, why?
5. I struggle to get my head around the MPC-rsfMRI gradient. I may have missed it, but I suggest adding more information on the biological meanings of the T1w/T2w ratio (e.g., Glasser et al., Nature 2016; Burt et al. Nat. Neuro. 2018). More critically, what does this gradient add to what we already know regarding the link between T1w/T2w and spatiotemporal dynamics (e.g., Demirtaş et al., Neuron, 2019)?
6. Because of the above-mentioned concerns, I am yet to be convinced about the evolutionary claims made in the discussion.

7. I suggest adding a sentence explaining the dual origin model of cortical differentiation before the Discussion.

8. Please add more details regarding the pre-processing of human replication fMRI data (p.24).

Reviewer #2 (Remarks to the Author):

In this paper, Valk et al. describe a study looking at the relationship between cortical structure and function and cognition. They used two different approaches. First, mapping and heritability estimation of structure-function relationships across the cortex in ~1,200 humans using data from the Human Connectome Project. Second, they compared the structure-function associations in humans with those seen in non-human primates, based on data from 19 macaques in the PRIMate-Data Exchange repository. The authors also performed additional analyses such as transcriptomic association using data from the Allen Human Brain Atlas and replication in a sample of 50 adults to assess the robustness of their findings.

Overall the methods are innovative, the paper is very well written and the research question is super interesting. The authors built microstructure profile covariance (MPC) matrices and resting-state functional connectivity (rsFC) matrices. Then, they correlated node-wise patterns in both measures, averaged across participants to find the overall edge-level association between MPC and rsFC. Relative to sensory/motor and unimodal areas, cortical "transmodal systems" showed the strongest 'decoupling' (lowest correlation between structure and function) between cortical microstructure and functional connectivity. The heritability analyses also indicated a lower heritability of transmodal cortical regions as determined using SOLAR (variance decomposition).

The paper is well written and the introduction and discussion do a good job contextualizing their work. My main criticism is the large number of panels in each of the figures. I wonder if some of those figures could go in supplementary materials or be separated into more figures. Adding a flowchart summarizing the methodological approach (either as main or supplementary) would also be helpful to readers.

Overall, very interesting paper.

Reviewer #3 (Remarks to the Author):

The authors present an analysis of the genetic, evolutionary and behavioral underpinnings/consequences of structure-function coupling in both humans and non-human primates. The analysis is thorough, replicating and contrasting human findings in non-human primates, and the results have the potential to impact our understanding of genetic control and evolution on structure function alignment. They also link their gradient analysis findings to brain function and make the case that decoupling of structure and function across evolution is greater in areas of the brain that underlie language and social cognition. There is much to like about this work and it has the potential for graet

impact on the field. However, I do have some concerns that, if addressed, would improve the robustness of and confidence in the results. I also believe that the transparency of the work could be greatly improved with the addition of a paragraph in the discussion outlining some of the limitations of the data/analysis, some of which are mentioned in my review below.

1) One of the biggest issues in my mind is that of noise in the MRI measurements. MRI (particularly fMRI) can be a noisy modality, and, if not accounted for or controlled for in some way, could corrupt the analysis. For example, in the heritability vs mean analysis - if a region or connection is noisy, it is likely to have both low structure-function coupling as well as low heritability, as noise will corrupt both associations. One way that measurement noise can be accounted for in heritability analysis would be to use an approach (Ge et al., 2017) that also accounts for within-subject variability (measurement noise) - this is able to be applied in HCP data where there are 4 fMRI scans per person (2 scans on 2 days) and measurement variability can be assessed. This approach was recently adapted for and applied to HCP data to quantify heritability of functional connectome fingerprints and structure-function coupling (here, structure refers to dMRI-based white matter connectivity). If the authors see fit, the latter paper could also be compared to the current paper's findings, as they represent a similar approach.

Ge, T., Holmes, A. J., Buckner, R. L., Smoller, J. W. & Sabuncu, M. R. Heritability analysis with repeat measurements and its application to resting-state functional connectivity. *Proc. Natl. Acad. Sci.*, 5521–5526, DOI: 10.1073/pnas.4771700765114 (2017).

Gu, Z., Jamison, K.W., Sabuncu, M.R. et al. Heritability and interindividual variability of regional structure-function coupling. *Nat Commun* 12, 4894 (2021). <https://doi.org/10.1038/s41467-021-25184-4>

2) In the same vein, it is important to assess test-retest reliability of MPC, rsFC and the coupling measure. This can be done with the subset of ~40 individuals in the HCP study that had a second MRI scan 6 months after the first. This will also help disentangle the effect of measurement noise on the various quantities (see my previous comment).

3) Apologies, but I am struggling to understand why assessing the correlation between the mean value of MPC/rsFC and their respective heritability estimates is helpful and how it can be interpreted (Figure 1D). Again, low coupling/MPC/rsFC being related to low heritability and high coupling/MPC/rsFC being related to high heritability may not indicate “genetic control” but could all be due to noise in the measures that impacts both quantities in the same direction. This needs to be controlled for (as suggested in my first comment) or at least assessed, possibly by showing an SNR map of the MRI measures from the test-retest analysis.

4) It is unclear how the node-wise measures in Figure 1D are being calculated.

5) Figure 1E is also a bit confusing to me, I am struggling to understand why the correlation between coupling and the correlation between the mean and heritability of MPC/rsFC is being assessed. It seems more interpretable and straightforward to instead look at the correlation between structure-function coupling and the two heritability estimates of MPC and rsFC. Then, it would support the current claim in

the paper:

“In other words, reduced structure-function coupling in human transmodal regions was paralleled by reductions in genetic control over both MPC and rsFC.”

My confusion could also be a result of my lack of understanding of the motivation to correlate mean and heritability values (see comment #3). It seems this correlation is being used to assess “genetic control” but isn’t that what heritability is providing? At minimum, please better motivate this part of the analysis and explain the reasoning in a more detailed manner.

6) The analysis with the macaques is an important one, but there are key differences in data acquisition that may be influencing the results. It is known that anesthesia can cause tighter coupling of structural connectivity and functional connectivity (Demertzi, 2019). At minimum, this should be considered in the discussion or by adding a limitations paragraph (which is currently not included, but would certainly help the reader fully understand the implications of the study).

Demertzi A, Tagliazucchi E, Dehaene S, et al. Human consciousness is supported by dynamic complex patterns of brain signal coordination. *Sci Adv.* 2019;5(2):eaat7603. doi:10.1126/sciadv.aat7603

Of particular note regarding this issue, in Figure 4A the higher-order fronto-parietal and DMN regions seem to have stronger structure-function coupling in macaques (upper left quadrant in the macaque panel). Further, the left-hand side of the quadrant that is dominated by rsFC seems to have stronger coupling in macaques. This could be due to the anesthesia causing a collapse of the rsFC to the anatomical networks, particularly for these higher-order regions that likely underlie consciousness.

7) Related to the above comment, it appears that all female macaques were used in the non-human primate data. Are there sex differences in coupling in the human data? If there do exist sex differences in structure-function coupling (in Gu, et al (2021), regionally-varying sex differences in structure-function coupling were in fact identified) then the comparison of human to non-human data could be biased. At minimum, an assessment of sex differences in coupling in humans should be provided and limitations in comparing these values to an all-female macaque dataset should be discussed.

8) Figure 3A could again be driven by the varying levels of noise, please see my previous comments for ways to try and account for or assess the influence of noise in both the MRI-based measures and the heritability measures. Please also clarify what is being shown in Figure 3A in the heatmaps in the bottom row.

9) Are there any covariates that should be considered in the rsFC, MPC or coupling analysis? In-scanner motion, age, sex, cognitive scores? In the discussion there is some mention of structure-function decoupling being linked to cognition as determined by the NeuroSynth analysis; there are cognitive scores for these subjects and this hypothesis could also be tested over the population of humans in the HCP study.

Minor comments:

- 1) What type of correlation was used to calculate the structure-function coupling measure (Pearson or Spearman) and was this choice justified by the shapes of the distributions? It states that in the heritability analysis, the FC and MPC values were transformed to adhere to normality assumptions; were the transformed values also used in the coupling calculation (if Pearson's correlation that assumes normality was used)?
- 2) Particularly in genetic analysis, one needs to determine if race is a driving factor. Race could be included as a co-variate in the heritability analysis or the heritability analysis should be reproduced using a homogenous group of Caucasian individuals.
- 3) In the transcriptomic analysis, it is stated that genes showing consistent whole-brain expression patterns across donors ($r > 0.5$) were retained; however, there is quite a sex imbalance in the Allen brain atlas data that could result in a bias toward only investigating genes that have higher expression rates in males. At a minimum, this limitation should be discussed.
- 4) The fourth paragraph of text in the Supplemental material section seems to be out of place?

For transparency, I will de-anonymize my review.

-Amy Kuceyeski

RESPONSE TO REVIEWERS (NCOMMS-21-25181-T)

We would like to thank the Editors and Reviewers for their positive evaluations, constructive comments, and for the opportunity to submit a revised manuscript. We feel that the comments and suggestions have greatly improved our manuscript. In this covering letter, we outline the steps we taken to address the suggestions of the Reviewers in a point-by-point fashion below and highlighted the corresponding changes in the manuscript in yellow.

REVIEWER #1 (R1; REMARKS TO THE AUTHOR):

This study combines heritability and cross-species (human and macaques) analyses to assess how changes in function-structure brain associations underpin cognition. The paper presents confirmatory findings (e.g., reduced function-structure coupling from sensory, unimodal, and transmodal regions) and some novel results (e.g., genetic control varies across the function-structure association axis; patterns of function-structure associations overlap in both species, with macaques showing a reduced decoupling in heteromodal but not paralimbic brain regions). Overall, the paper is well written, and its structure easy to follow, with figures that are clear and informative, and the methods state-of-the art. However, at this stage, I have the following concerns that seem to preclude an unequivocal interpretation of the key results:

We thank the Reviewer for the appreciation of our work and the insightful comments, which we have addressed below.

1. The increased function-structure association in macaques relative to humans is an interesting finding. However, I have the following worries:

a. Could this result be explained by differences in the quality of data between the species (e.g., better signal to noise ratio in macaque data compared to humans)?

We agree with the Reviewer that differences in signal to noise may contribute to cross-species differences. Indeed, tSNR was higher in non-human primates (NHP: Davis: mean \pm SD: 804.48 \pm 206.34). Similar tSNR was found in two supporting macaque samples that we additionally evaluated as part of this revision including male and female with a different age-range, and awake as well as anesthetized monkeys (Newcastle (awake): 282.31 \pm 86.05; Oxford (anesthetized): 673.61 \pm 209.115), compared to the human sample (HCP: 228.31 \pm 54.24). Mapping human tSNR to macaque space via a previously established procedure (T. Xu et al., 2020) and computing difference per region, tSNR was consistently increased in NHP relative to humans, in particular in visual cortex. Importantly, however, we did not observe a marked spatial correlation of the tSNR maps of both species (Spearman's $r=0.03$, $p>0.1$), nor a correlation between the difference in tSNR and the difference structure-function coupling between species (Spearman's $r=0.00$, $p>0.1$). This suggests that differences in tSNR likely do not contribute to cross-species differences in structure-function coupling, as then we would expect to observe spatial relationships between differences in tSNR and uncoupling. See *Results*, P.9.

“Differences between humans and macaques did not show a significant relation with differences in tSNR.”

Supplementary Results, P.3

“Overall, tSNR varied between species, with macaques showing higher tSNR than humans. When mapping human tSNR to macaque space (T. Xu et al., 2020) and computing the difference per region, we also found widespread increases in tSNR in macaques compared to humans across all regions, in particular in visual cortex. However, correlating tSNR between species, we observed no spatial relationship (Spearman's $r=0.03$, $p>0.1$) nor a relationship between difference in tSNR and difference in coupling between species (Spearman's

$r=0.00$, $p>0.1$). This suggests that the observed uncoupling of MPC and rsFC in humans cannot be accounted for by variations in tSNR alone (**Supplementary Fig. 6**).”

Supplementary Fig 6. tSNR in humans and macaque samples. A-D). Cortical tSNR (mean/SD) in the main human (HCP) and macaque sample (Davis, anesthetized) as well as two additional macaque samples (Newcastle, awake; Oxford, anesthetized); **E)** Difference in tSNR between humans mapped to macaque space (HCP) and macaques (Davis, main analysis sample), their correlation (left lower scatterplot), and correlation of difference in tSNR and structure-function coupling differences (right lower scatterplot).

Methods, P. 22:

“To evaluate robustness of the macaque findings, we evaluated two additional macaque samples with rsFC data openly available in PRIME-DE (http://fcon_1000.projects.nitrc.org/indi/indiPRIME.html) (Milham et al., 2018), previously used in (T. Xu et al., 2020). The samples varied in age-range, sex, and acquisition parameters. The Newcastle dataset (awake (Rinne, Muers, Salo, Slater, & Petkov, 2017; Schonwiesner, Dechent, Voit, Petkov, & Krumbholz, 2015)) consisted of 10 rhesus macaques (8 males, age mean \pm SD=8.28 \pm 2.33, weight=11.76 \pm 3.38) scanned on a Vertical Bruker 4.7T primate scanner. The fMRI session was acquired with 1.2 \times 1.2 \times 1.2 mm resolution, TR=2,000 ms, 8.33-min per scan (250 volumes \times 2 scan) per animal. No contrast-agent was used during the scans. In the case of the Oxford dataset (anesthetized), we included nineteen rhesus macaques with preprocessing and surface reconstruction as in previous work (T. Xu et al., 2020)(all males, age=4.01 \pm 0.98 years, weight=6.61 \pm 2.04 kg). The macaques were scanned on a 3T with a 4-channel coil (Noonan et al., 2014). Resting-state fMRI (rs-fMRI) data were collected with 2 mm isotropic resolution, TR=2,000 ms, 53.3 min (1,600 volumes). No contrast-agent was used during the scans.”

b. Could the suppressed brain activity (via anesthesia) in macaques explain the result (e.g., increased mapping of function on structure due to reduced variability)?

We selected the Davis sample of NHP for our primary analysis, as this sample was the only macaque sample with high resolution T1wT2w MRI available in combination with rsFC. However, we agree that it is possible that the anesthetized nature of the animals in this sample could have contributed to our findings. To assess effects of anesthesia on structure-function coupling, we conducted a supplementary analysis in which we explored the associations

between the Davis microstructure and rsFC measures from two additional macaque samples: a sample of awake macaques (Newcastle; N=10, 8 males, age=8.28±2.33 yrs), as well as an additional sample of anesthetized young macaque males (Oxford; N=19 males, age=4.01±0.98 yrs; see comment #1C below). Despite some inter-site differences, patterns of decoupling were comparable for anesthetized and awake samples (Davis-Newcastle: $r=0.78$; Davis-Oxford: $r=0.90$; Newcastle-Oxford: $r=0.71$). All samples showed correlation with the human HCP measure of structure-function coupling (Davis: Spearman's $r=0.48$, $p_{\text{spin}}=0.016$; Newcastle: Spearman's $r=0.36$, $p_{\text{spin}}=0.06$; Oxford: Spearman's $r=0.40$, $p_{\text{spin}}=0.05$). Similar to our main findings, we observed high coupling for all macaque samples in idiotypic, unimodal, and heteromodal regions, whereas only paralimbic regions were uncoupled. Conversely, in human data we observed relative uncoupling in unimodal, heteromodal, and paralimbic regions.

Sample	Idiotypic [mean(SD)]	Unimodal	Heteromodal	Paralimbic
HCP	0.465(0.181)	0.149(0.263)	0.063(0.198)	0.012(0.146)
Davis	0.376(0.260)	0.346(0.221)	0.333(0.255)	0.052(0.230)
Newcastle	0.322(0.167)	0.262(0.156)	0.249(0.222)	0.108(0.245)
Oxford	0.433(0.255)	0.390(0.224)	0.403(0.269)	0.166(0.284)

Supplementary Table 2. Structure-function coupling mean(SD) as a function of cytoarchitectural class in humans and macaques.

Sample	Idiotypic [mean(SD)]	Unimodal	Heteromodal	Paralimbic
Davis- HCP	0.0628 ($p>0.1$)	6.399 ($p<0.001$)	6.445 ($p<0.001$)	0.063 ($p>0.1$)
Newcastle - HCP	-0.6713 ($p>0.1$)	4.516 ($p<0.001$)	4.759 ($p<0.001$)	2.184 ($p>0.1$)
Oxford - HCP	0.699 ($p>0.1$)	7.744 ($p<0.001$)	7.905 ($p<0.001$)	3.157 ($p=0.01$)

Supplementary Table 3. Difference in structure-function coupling between humans and macaque across samples

Supplementary Fig 7. Structure-function coupling in different macaque samples. A) Anesthetized macaque (sample: Davis) structure-function coupling and correlation with human structure-function coupling projected to macaque space; B) Awake macaque (sample Newcastle) structure-function coupling and correlation with human structure-function coupling projected to macaque space; C) Anesthetized macaque (sample Oxford) structure-function coupling and correlation with human structure-function coupling projected to macaque space; D) Structure-function coupling averaged in cytoarchitectural classes (M. Mesulam, 2000).

We included the table and figure in the *Supplementary Results*.

Results, P. 9:

“Replacing the macaque rsFC matrix with two other samples, including males and females with a different age-range, and awake as well as anesthetized monkeys, yielded broadly similar patterns of structure-function coupling (Supplementary Results).”

Supplementary Results, P. 3:

*“To evaluate whether the observed structure-function coupling in macaques might be due to awake versus anesthetized status, as well as differences in age-range and sex between samples, we evaluated two additional macaque samples, the Newcastle and Oxford sample (**Supplementary Fig. 7**). As these datasets did not have high-resolution microstructural MRI data available, we correlated the region-wise average functional connectivity data of the two additional samples, thought to be most impacted by anesthesia, with the microstructural profile covariance data of the Davis sample. Despite some inter-site differences, patterns of decoupling were similar for anesthetized and awake samples with different sex and age distributions (Davis-Newcastle: $r=0.78$; Davis-Oxford: $r=0.90$; Newcastle-Oxford: $r=0.71$). All samples showed correlation with the human HCP measure of structure-function coupling (Davis: Spearman’s $r=0.48$, $p_{spin}=0.012$; Newcastle: Spearman’s $r=0.36$, $p_{spin}=0.06$; Oxford: Spearman’s $r=0.40$, $p_{spin}=0.05$). For all macaque samples, we observed similarly high coupling in idiosyncratic, unimodal and heteromodal regions, whereas only paralimbic regions were uncoupled. Conversely, in human data we observed relative uncoupling in unimodal, heteromodal and paralimbic regions (**Supplementary Table 2 and 3**).*

Discussion, P. 17/18:

“Moreover, previous work has indicated differences in brain function at rest relate to the level of consciousness in humans and macaques (Demertzi et al., 2019; T. Xu et al., 2019), and also our findings suggest subtle differences in coupling of structure and function between awake and anesthetized macaques. Despite this variation, we observed a consistent difference in unimodal and heteromodal coupling between (awake) humans and awake as well as anesthetized macaques. Future work may expand on this work to further explore variations in structure-function coupling by level of consciousness, particularly in humans by exploiting the capacity for introspection. Such analysis may provide additional insight in the relation between structure-function coupling and features of internal cognition.”

c. Are human and macaque data age-equivalent? If not, how are different neurodevelopmental states accounted for?

The primary non-human sample used in this study, Davis, consisted of middle-age females and was selected as the only sample that had high-resolution microstructural and rsFC data available, enabling comparisons with human analysis. However, when additionally assessing other macaque samples with different age (and sex) distributions (Davis: anesthetized, N=19, all female, age=20.38±0.93 yrs; Newcastle: awake, N=10, 8 males, age=8.28±2.33 yrs; Oxford: anesthetized, N=19, all male, age=4.01±0.98 yrs), we could show similar structure-function coupling (*Supplementary Figure 7* and *Supplementary Table 3*). We added this information to the *Supplements* and *main Results*, please also see R1#1b for in-text additions.

2. Expanding on my previous comment: The pre-processing of human and macaques resting-state data appears quite different- e.g., ICA FIX in humans, temporal filter and smoothing in macaques. The fact that a number of results match across species despite these differences can be seen as a strength, but such differences make the interpretation of across-species differences problematic. Replicating the analyses with different data from a different primate species (e.g., marmoset; Liu C. et al. Nature Comm., 2019, perhaps adopting a proxy measure of T1w/T2w if not available) could help to address this concern. The authors may come up with other confirmatory approaches to address the concern.

To address this important concern, we conducted additional analysis including two new macaque samples, including male and female with a different age-range, and awake as well as anesthetized monkeys (see previous comments). Moreover, we conducted a supplementary analysis that examined structure-function in another human sample (eNKI) that underwent the same processing as the macaque data. All these analyses highlight the same pattern of uncoupling of microstructure and function across samples, and increased uncoupling in unimodal and heteromodal regions in macaque samples relative to human samples.

Specifically, we examined structure-function coupling by replacing the resting-state correlation matrix of the HCP sample with a macaque sample-equivalent functional data (based on another human sample (eNKI)) that underwent the same rsFC processing as the macaque data, i.e. temporal compression, motion correction, 4D global scaling, nuisance regression (Friston's 24 model, cerebrospinal fluid and white matter), linear and quadratic detrends, band-pass filtering (0.01–0.1 Hz), and surface registration (for further details, see (Nenning et al., 2020)). We again observed increased uncoupling of unimodal and heteromodal, but not idiosyncratic and paralimbic, regions in humans relative to macaques (idiosyncratic $t=-1.199$, $p>0.1$; unimodal $t=5.161$, $p<0.001$; heteromodal $t=5.690$, $p<0.001$; paralimbic: $t=-2.182$, $p>0.1$).

Sample	Idiosyncratic [mean(SD)]	Unimodal	Heteromodal	Paralimbic
HCP	0.465(0.181)	0.149(0.263)	0.063(0.198)	0.012(0.146)
eNKI	0.431(0.131)	0.176(0.230)	0.134(0.202)	0.144(0.170)

Supplementary Figure 9. Structure-function coupling in dataset processed comparable to the rsFC of the macaque sample

Please see the revised *Methods*, P.27.

“To evaluate whether differences in rs-fMRI pre-processing between humans and macaques can explain differences observed in coupling of MPC and rsFC between both species, we evaluated structure-function coupling in an additional human sample (eNKI subsample. N=100, age-range 18-40yrs) preprocessed analogously to the macaque datasets, including temporal compression, motion correction, 4D global scaling, nuisance regression (Friston's 24 model, cerebrospinal fluid and white matter), linear and quadratic detrends, band-pass filtering (0.01–0.1 Hz), and surface registration (for details, see (Nenning et al., 2020)).

And *Results*, P.9.

“Moreover, structure-function coupling in humans was reproducible in another human sample (eNKI; N=100, age=18-40yrs) preprocessed analogous to the macaque sample ($r=0.94$ with structure-function coupling in HCP, difference between humans and macaques as a function of cytoarchitectural class: idiosyncratic $t=-1.199$, $p>0.1$; unimodal $t=5.161$, $p<0.001$; heteromodal $t=5.690$, $p<0.001$; paralimbic: $t=-2.182$, $p>0.1$), suggesting differences between humans and macaques went beyond differences in preprocessing (Supplementary Results).”

We agree that showing consistency of microstructure-function coupling across different primate species, beyond the replication in other macaque samples (R1.1b), will be key to properly understand the generalizability of structure-function coupling in the mammalian

brain, particularly also uncoupling observed in unimodal and heteromodal regions. Anatomical differences in these regions, including frontal and parietal cortices, of other primates relative to the humans have been reported, including differences in cortical and deep white matter architecture (Donahue, Glasser, Preuss, Rilling, & Van Essen, 2018; Eichert et al., 2020; Rilling & Insel, 1999). Moreover, while previous functional work has reported homologs of the DMN, a key functional network situated in heteromodal areas, in other primates beyond macaques (T. Xu et al., 2020), including marmosets (C. Liu et al., 2019) and chimpanzees (Barks, Parr, & Rilling, 2015), this organization has shown to diverge between humans and macaques (T. Xu et al., 2020). It is possible that anatomical differences underlie divergences in functional organization of heteromodal regions, including the DMN, between humans and non-human primates (C. Liu et al., 2019; Schaeffer et al., 2020; T. Xu et al., 2020), which in turn may be reflected in reduced uncoupling of structure and function in these regions in macaques.

In the current study, we were able to compare humans and macaques directly using cross-species alignment techniques (T. Xu et al., 2020), in combination with histologically-validated *in vivo* microstructure profile covariance maps (Paquola, Vos De Wael, et al., 2019). Both the used alignment techniques and microstructural profile covariance mapping are so far only feasible in humans and macaques using comparable *in vivo* data in both species. Nevertheless, our observations comparing structure-function coupling and organization suggest a pattern of uncoupling of transmodal regions is phylogenetically conserved in the two primate species under study, ruling out that this particular cortical motif is present only in humans. Thus, while it is an important question to address precisely when during evolution this pattern emerged, it is beyond the purpose of our current analysis. However, as we agree with the Reviewer that this is an important question, we consider how this question will likely to become increasingly tractable over the next few years as more *in vivo* datasets will become available in non-human primates that make further comparisons possible (Hayashi et al., 2021; Milham et al., 2018). We added this point, and limitations of the evolutionary analysis to the *Discussion*, P.17.

“In the current study, we report genetic uncoupling in transmodal regions in humans using a twin-design and differences between humans and macaques in terms of structure-function associations. We were able to compare humans and macaques directly using previously established cross-species alignment techniques (T. Xu et al., 2020) in combination with histologically validated in vivo microstructure profile covariance analysis (Paquola, Vos De Wael, et al., 2019). As it stands, meaningful comparisons between species using the current techniques are limited to these two species. Thus, although we could show that the patterning of structure-function uncoupling is consistent in humans and macaques, and therefore, likely important in many primates, we can only speculate about the evolutionary relevance of the observed differential uncoupling in unimodal and heteromodal regions across species. Anatomical differences in these regions, including frontal and parietal cortices, of other primates relative to humans have been reported, including differences in cortical and deep white matter architecture (Donahue et al., 2018; Eichert et al., 2020; Rilling & Insel, 1999). Though previous functional work has reported homologs of the default mode network in other primates beyond macaques (T. Xu et al., 2020), including marmosets (C. Liu et al., 2019) and chimpanzees (Barks et al., 2015), it has recently been reported that the functional network hierarchy is different in humans relative to macaques, with regions in the default mode regions showing most divergence (T. Xu et al., 2020). One possible hypothesis is that there is an evolutionary adaptation in structure and function of unimodal and heteromodal association regions between humans and NHPs (C. Liu et al., 2019; Schaeffer et al., 2020; T. Xu et al., 2020), resulting in progressive untethering of structure and function in the former that gives rise to aspects of cognition that are most developed in humans (Buckner & Krienen, 2013).”

And *Discussion*, P.18.

“To test this hypothesis, future work should expand our approach to include a wider range of species, and explore their association with different functional states in a more detailed manner. These studies will be an important step towards the understanding of how evolution has shaped neural function to support important

features of human cognition, a research goal which is likely to become increasingly feasible with growing availability of open multispecies in vivo datasets (Hayashi et al., 2021; C. Liu et al., 2019; Milham et al., 2018).”

3. Are the findings robust against differences in individual-specific data and brain volume (e.g., coupling-brain volume)?

This is an important consideration for robustness of our results and as suggested by the Reviewer, we reran structure-function analysis in individual human participants, averaged the maps, and obtained a high correspondence to the group-average structure-function findings ($r=0.72$, range [0.38 0.86]; Supplementary Fig 10). Correlating individual variation in structure-function coupling with ICV, individuals with higher ICV (controlling of age and sex) showed stronger coupling in left superior frontal and right posterior insular regions ($FDRq<0.05$) and decreased left posterior temporal coupling ($FDRq<0.05$). However, associations between ICV and structure-function coupling did not correlate with the pattern of structure-function coupling itself ($r=0.12$, $p_{spin}>0.1$). Combined, these analyses indicate similar results between individual and group-level data and robustness relative to ICV variations in humans. See *Supplementary Results*:

“Probing inter-individual variation of microstructure-function coupling in humans, we observed a high correspondence between single-subject and group-average findings (mean r : 0.72 range: [0.38 0.86]), indicating consistency. Also evaluating potential effects of ICV on structure-function coupling, we observed positive and negative associations between coupling and ICV (stronger coupling in left superior frontal and right posterior insular regions ($FDRq<0.05$) and decreased coupling in right posterior temporal regions ($FDRq<0.05$)). Importantly, however, there was only a weak and non-significant link with mean patterns of structure-function coupling ($r=0.11$, $p_{spin}>0.1$). Combined, these analyses suggest that our results are robust at the level of the individual and against variations in ICV.”

Supplementary Fig 10. Individual variation in structure-function coupling. A) Mean MPC-rsFC coupling across participants; B) Association between individual MPC-rsFC maps and mean MPC-rsFC map; C) Correlation between MPC-rsFC coupling and intracranial volume per parcel and averaged within cytoarchitectural class (M. Mesulam, 2000).

4. Gradient decomposition: How robust are the results to changes in the parameters alpha and t? Were these parameters identical for the analysis of human and macaque connectomes? If not, why?

Apologies for the unclarity. Parameters were identical for humans and macaques. See *Methods*, P. 24.

“Gradient parameters were identical for human and macaque data.”

Gradient findings were highly consistent when varying α and t and both MPC and rsFC gradients could be reproduced in humans and macaques at $r > 0.99$ when varying α between 0 and 1 (standard value 0.5), and diffusion time between 0 and 10 (standard value 0). We added this information to the *Supplementary Results, P. 3*.

“Computing robustness of gradient as a function of varying α [0 1] and diffusion time [0 9] we found gradients to be highly similar ($r > 0.99$) to each other in human and macaque data.”

And *Methods, P. 24*

“To evaluate robustness of the gradient measures we varied α between 0 and 1 with steps of 0.1 and diffusion time between 0 and 9 with steps of 1 in humans and macaque rsFC and MPC data.”

Supplementary Fig 8. Intercorrelation of gradients as a function of varying α and diffusion-time. **A)** Varying α and diffusion time for rsFC gradient in humans; **B)** Varying α and diffusion time for MPC gradient in humans; **C)** Varying α and diffusion time for rsFC gradient in macaques; and **D)** Varying α and diffusion time for MPC gradient in macaques. The parameters used for main analyses in humans and macaques in the current study are 0.5 α and 0 diffusion time, similar to previous work (Margulies et al., 2016; Paquola, Vos De Wael, et al., 2019).

5. I struggle to get my head around the MPC-rsfMRI gradient. I may have missed it, but I suggest adding more information on the biological meanings of the T1w/T2w ratio (e.g., Glasser et al., Nature 2016; Burt et al. Nat. Neuro. 2018). More critically, what does this gradient add to what we already know regarding the link between T1w/T2w and spatiotemporal dynamics (e.g., Demirtaş et al., Neuron, 2019)?

T1w/T2w has previously been suggested to relate to differential myelination of the cerebral cortex (Assem, Glasser, Van Essen, & Duncan, 2020; Ganzetti, Wenderoth, & Mantini, 2014; Glasser et al., 2016; Glasser & Van Essen, 2011). However, there is no 1:1 mapping and these MRI-based measures are also sensitive to signal changes due to iron and water, as well as other cytological features including dendritic arborization, cell size, and cell density (Burt et al., 2018; Lorio et al., 2016; Stuber et al., 2014). We now included this in the *Introduction, P.5*;

“The T1w/T2w ratio has been shown to reflect myelin content (Assem et al., 2020; Ganzetti et al., 2014; Glasser et al., 2016; Glasser & Van Essen, 2011) but also iron, water, as well as cytological variations including dendritic arborization, cell size, and cell density (Burt et al., 2018; Lorio et al., 2016; Stuber et al., 2014).”

Cortex-wide variation of mean T1w/T2w values have been shown to reflect hierarchical differentiation of cortical areas, with lower myelin content in areas with neural dynamics occurring at slower dynamic time scales that are thought to be involved in more integrative and higher-order processing in both humans (Burt et al., 2018; Demirtas et al., 2019) and other primates (Murray et al., 2014).

The MPC approach, and associated principal gradient, complements T1w/T2w ratio mapping, by focusing on structural variations across cortical depths similar to approaches formulated in prior neuroanatomy studies (Schleicher, Amunts, Geyer, Morosan, & Zilles, 1999; Zilles et al., 2002). Depth-dependent shifts in cellular and myelin characteristics have been shown to reflect architectural complexity (Zilles et al., 2002) and cortical hierarchy (M. M. Mesulam, 1998). Thus, despite similarities in the gradients derived from mean T1w/T2w and MPC analysis (Paquola, Bethlehem, et al., 2019), approaches and the resulting patterns are not equivalent. From a practical perspective, depth-dependent profiling enables the computation of inter-regional microstructural profile covariance (MPC) networks in single subjects. In prior work, we showed that MPC gradients based on T1w/T2w (but also qT1 and MT MRI) show a sensory-fugal gradient that corresponds to microstructural profiles derived from sub-millimeter 3D *post mortem* histology based on the BigBrain dataset (Paquola, Bethlehem, et al., 2019; Paquola, Vos De Wael, et al., 2019). Thus, MPC provides information on depth-specific local and inter-regional features and dependencies, that are not captured by mean T1wT2w.

As the goal of our study was to investigate how the inter-regional similarity of microstructural profiles relates to functional connectivity, we compared, amongst others, the principal organizational gradients of both measures, to investigate whether main axes of microstructural covariance and functional connectivity were similar or distinct. This is a different objective/question to the one presented by Demirtas and colleagues, who investigated whether a proxy of cortical hierarchy (based on average T1wT2w) could enhance microcircuit models. Despite this difference in local versus inter-regional approaches, our findings nevertheless converge with the work of Demirtas et al., as we show an increased uncoupling of MPC and rsFC from idiosyncratic to paralimbic regions, suggesting a different association between structure and function along this hierarchical axis. We now further differentiated T1w/T2w-based MPC from mean T1w/T2w mapping in the *Methods*, P. 20:

“The MPC approach complements mean T1w/T2w ratio mapping, tapping into microstructural profiles across cortical depths. Depth-dependent cortical microstructure analysis has a long tradition in neuroanatomy (Schleicher et al., 1999; Zilles et al., 2002), and depth-dependent shift in cellular and myelin characteristics have been shown to reflect architectural complexity (Zilles et al., 2002) and cortical hierarchy (M. M. Mesulam, 1998).”

Moreover, we contextualized the study of Demirtas et al. in our *Discussion*, P.15:

“At the same time, microcircuit models of cortical dynamics have been shown to benefit from integrating regionally-varying myelin sensitive MRI information, highlighting that cortical dynamics vary as a function of local microstructural context (Demirtas et al., 2019). This indicates a different coupling of microstructure and intrinsic function along the cortical hierarchy.”

6. Because of the above-mentioned concerns, I am yet to be convinced about the evolutionary claims made in the discussion.

We hope the clarifications and further analyses outlined above have addressed the specific concerns. Our data suggests that there are similarities between humans and macaques, indicating the transmodal uncoupling of microstructure and function is a process that is shared among at least two primate species. Though studying this neural motif across a broader range of primate species will aid to fully understand the evolutionary basis of this pattern, our existing analysis is sufficient to support the claim that this basic process is not unique to humans. At the same time, we observed stronger uncoupling in unimodal and heteromodal regions in humans relative to macaques, which was paralleled by decreases in heritability and genetic coupling, in particular in heteromodal regions (see also R3Q1). The observed similarities and differences may reflect more general phylogenetic patterns, but could also be specific to humans and macaques. We now further specified our interpretation in the *Discussion* to highlight these differences between humans and macaques

P.15.

“Notably, however, regions in which structure-function decoupling showed differences between human and macaques, and differences between structural and functional organization, were associated with functions such as “social-cognition” and “episodic/autobiographical memory”.”

P.16.

“Together these observations suggest that differentiable genetic processes may underlie transmodal uncoupling in heteromodal and paralimbic cortices, an important hypothesis for future studies to examine.”

Additionally, we agree that comparisons across the broader phylogenetic tree including more primates will provide more comprehensive information about potential evolutionary processes, rather than comparisons between two species (Barton & Venditti, 2013; Heuer et al., 2019). To ensure that this important distinction is clear to the reader we have now explicitly considered this in the *Discussion* highlighting potential other factors to consider when comparing across species, P.17 and P.18.

“In the current study, we report genetic uncoupling in transmodal regions in humans using a twin-design and differences between humans and macaques in terms of structure-function associations. We were able to compare humans and macaques directly using previously established cross-species alignment techniques (T. Xu et al., 2020) in combination with histologically validated in vivo microstructure profile covariance analysis (Paquola, Vos De Wael, et al., 2019). As it stands, meaningful comparisons between species using the current techniques are limited to these two species. Thus, although we could show that the patterning of structure-function uncoupling is consistent in humans and macaques, and therefore, likely important in many primates, we can only speculate about the evolutionary relevance of the observed differential uncoupling in unimodal and heteromodal regions across species. Anatomical differences in these regions, including frontal and parietal cortices, of other primates relative to humans have been reported, including differences in cortical and deep white matter architecture (Donahue et al., 2018; Eichert et al., 2020; Rilling & Insel, 1999). Though previous functional work has reported homologs of the default mode network in other primates beyond macaques (T. Xu et al., 2020), including marmosets (C. Liu et al., 2019) and chimpanzees (Barks et al., 2015), it has recently been reported that the functional network hierarchy is different in humans relative to macaques, with regions in the default mode regions showing most divergence (T. Xu et al., 2020). One possible hypothesis is that there is an evolutionary adaptation in structure and function of unimodal and heteromodal association regions between humans and NHPs (C. Liu et al., 2019; Schaeffer et al., 2020; T. Xu et al., 2020), resulting in progressive untethering of structure and function in the former that gives rise to aspects of cognition that are most developed in humans (Buckner & Krienen, 2013).”

“To test this hypothesis, future work should expand our approach to include a wider range of species, and explore their association with different functional states in a more detailed manner. These studies will be an important step towards understanding how evolution has shaped neural function to support important features of human cognition, a research goal which is likely to become increasingly feasible with growing availability of open multispecies in vivo datasets (Hayashi et al., 2021; C. Liu et al., 2019; Milham et al., 2018).”

7. I suggest adding a sentence explaining the dual origin model of cortical differentiation before the Discussion.

We thank the Reviewer and now describe the dual origin model in the *Introduction* on P.5:

“In particular, we evaluated the relationship between structure-function coupling with evolutionary axes functional re-organization (T. Xu et al., 2020), and the dual origin model (Dart, 1934; Pandya, Petrides, Seltzer, & Cipolloni, 2015). The latter model assumes that cortical areas develop from waves of laminar differentiation that have their origin in either the piriform cortex (paleocortex) or the hippocampus (archicortex) (Pandya et al., 2015; Sanides, 1962).”

8. Please add more details regarding the pre-processing of human replication fMRI data (p.24).

We are happy to provide further information to the resting-state fMRI replication data processing in the revised *Methods*, P. 27:

“As previously described (Royer & al., biorXiv), rs-fMRI data were pre-processed using a combination of AFNI (Cox, 1996) and FSL (Jenkinson, Beckmann, Behrens, Woolrich, & Smith, 2012). To ensure magnetic field saturation, the first five volumes were disregarded. Images were reoriented, motion and distortion corrected. Motion correction was performed by registering all timepoints to the mean volume, while distortion correction leveraged main phase and reverse phase field maps acquired alongside rs-fMRI scans. Nuisance variable signal was removed using an in-house trained ICA-FIX classifier (Salimi-Khorshidi et al., 2014) and via spike regression using motion outlier outputs provided by FSL. Volumetric timeseries were averaged for registration to native FreeSurfer space using boundary-based registration (Greve & Fischl, 2009), and mapped to individual surface models using trilinear interpolation. Native surface cortical timeseries underwent spatial smoothing once mapped to each individual’s cortical surface models (Gaussian kernel, FWHM=10mm) (Glasser et al., 2013; Marcus et al., 2011), and were subsequently averaged within nodes defined by the Schaefer 400 parcellation.”

REVIEWER #2 (REMARKS TO THE AUTHOR):

In this paper, Valk et al. describe a study looking at the relationship between cortical structure and function and cognition. They used two different approaches. First, mapping and heritability estimation of structure-function relationships across the cortex in ~1,200 humans using data from the Human Connectome Project. Second, they compared the structure-function associations in humans with those seen in non-human primates, based on data from 19 macaques in the PRIMate-Data Exchange repository. The authors also performed additional analyses such as transcriptomic association using data from the Allen Human Brain Atlas and replication in a sample of 50 adults to assess the robustness of their findings.

Overall the methods are innovative, the paper is very well written and the research question is super interesting. The authors built microstructure profile covariance (MPC) matrices and resting-state functional connectivity (rsFC) matrices. Then, they correlated node-wise patterns in both measures, averaged across participants to find the overall edge-level association between MPC and rsFC. Relative to sensory/motor and unimodal areas, cortical "transmodal systems" showed the strongest 'decoupling' (lowest correlation between structure and function) between cortical microstructure and functional connectivity. The heritability analyses also indicated a lower heritability of transmodal cortical regions as determined using SOLAR (variance decomposition).

The paper is well written and the introduction and discussion do a good job contextualizing their work. My main criticism is the large number of panels in each of the figures. I wonder if some of those figures could go in supplementary materials or be separated into more figures. Adding a flowchart summarizing the methodological approach (either as main or supplementary) would also be helpful to readers.

Overall, very interesting paper.

We thank the Reviewer for the positive evaluation and the helpful suggestions. As suggested, we split Fig. 1 in two figures and moved the upper panel (gradients of heritability of rsFC/MPC) of Fig. 3 to the Supplements. A flowchart summarizing the approach was added to the *Supplementary Materials*, and referred to in the main manuscript on P. 5.

*“See **Supplementary Figure 1** for schematic of analyses.”*

Fig 1. Coupling of microstructural profile covariance (MPC) and functional connectivity at rest (rsFC).

Fig 2. Genetic coupling of microstructural profile covariance (MPC) and functional connectivity at rest (rsFC).

Fig 3. Organizational difference between MPC and rsFC in humans and macaques

Updated Supplementary Fig 4.

Supplementary Fig 1. Analysis scheme. Analysis scheme colored by question and analysis type.

REVIEWER #3 (REMARKS TO THE AUTHOR):

The authors present an analysis of the genetic, evolutionary and behavioral underpinnings/consequences of structure-function coupling in both humans and non-human primates. The analysis is thorough, replicating and contrasting human findings in non-human primates, and the results have the potential to impact our understanding of genetic control and evolution on structure function alignment. They also link their gradient analysis findings to brain function and make the case that decoupling of structure and function across evolution is greater in areas of the brain that underlie language and social cognition. There is much to like about this work and it has the potential for great impact on the field. However, I do have some concerns that, if addressed, would improve the robustness of and confidence in the results. I also believe that the transparency of the work could be greatly improved with the addition of a paragraph in the discussion outlining some of the limitations of the data/analysis, some of which are mentioned in my review below.

We thank the Reviewer for the appreciation of the work and the helpful comments, which we have addressed below.

1) One of the biggest issues in my mind is that of noise in the MRI measurements. MRI (particularly fMRI) can be a noisy modality, and, if not accounted for or controlled for in some way, could corrupt the analysis. For example, in the heritability vs mean analysis - if a region or connection is noisy, it is likely to have both low structure-function coupling as well as low heritability, as noise will corrupt both associations. One way that measurement noise can be accounted for in heritability analysis would be to use an approach (Ge et al., 2017) that also accounts for within-subject variability (measurement noise) - this is able to be applied in HCP data where there are 4 fMRI scans per person (2 scans on 2 days) and measurement variability can be assessed. This approach was recently adapted for and applied to HCP data to quantify heritability of functional connectome fingerprints and structure-function coupling (here, structure refers to dMRI-based white matter connectivity).

If the authors see fit, the latter paper could also be compared to the current paper's findings, as they represent a similar approach.

*Ge, T., Holmes, A. J., Buckner, R. L., Smoller, J. W. & Sabuncu, M. R. Heritability analysis with repeat measurements and its application to resting-state functional connectivity. *Proc. Natl. Acad. Sci.*, 5521–5526, DOI: 10.1073/pnas.4771700765114 (2017).*

*Gu, Z., Jamison, K.W., Sabuncu, M.R. et al. Heritability and interindividual variability of regional structure-function coupling. *Nat Commun* 12, 4894 (2021). <https://doi.org/10.1038/s41467-021-25184-4>*

We thank the Reviewer for this suggestion and incorporated the approach from Ge et al. 2017 and Gu et al. 2021 to assess intra-individual variation. Overall, using this approach we were able to replicate previous patterns and in particular the observed patterns of increased heritable coupling in idiosyncratic (mean±std Spearman's $r=0.789\pm 0.121$) and unimodal (Spearman's $r=0.704\pm 0.162$) regions, whereas coupling in heteromodal (Spearman's $r=0.580\pm 0.175$) and paralimbic (Spearman's $r=0.433\pm 0.245$) regions was reduced. These patterns correlated with MPC-rsFC coupling (Spearman's $r=0.416$, $p_{\text{spin}}=0.01$) and Solar-based heritability metrics (Spearman's $r=0.754$, $p<0.001$). Of note, although heritability values increase overall, paralimbic regions, in particular, show marked reductions relative to Solar-based heritability (Supplementary Figure 14). Such differences may reflect reductions

of data quality due to reduced scanning time. We have now included this alternative computation of heritability in the *Supplementary Materials*, P.5.

“Computing heritability using an alternative measure suggested to account for intra-individual variation (Ge, Holmes, Buckner, Smoller, & Sabuncu, 2017), we observed overall consistent patterns. Specifically, we observed strong heritable coupling in idiotypic (Spearman’s $r=0.789\pm0.121$) and unimodal (Spearman’s $r=0.704\pm0.162$) regions, whereas coupling in heteromodal (Spearman’s $r=0.580\pm0.175$) and paralimbic (Spearman’s $r=0.433\pm0.245$) regions was reduced. These patterns correlated with MPC-rsFC coupling (Spearman’s $r=0.416$, $p_{spin}=0.01$) and Solar-based heritability metrics (Spearman’s $r=0.754$, $p<0.001$).”

And *Methods*, P. 22.

“To evaluate robustness of our heritability findings of rsFC with respect to intra-individual variation we additionally computed heritability considering possible within-subject variation described in previous work (Ge et al., 2017; Gu, Jamison, Sabuncu, & Kuceyeski, 2021).”

Supplementary Fig 14. rsFC – rsFC heritability coupling considering potential random effects. Scale r_{ranked} [-1 1]; distribution of rsFC-rsFC h2 coupling within cytoarchitectural classes; scatter of MPC-rsFC coupling versus rsFC – rsFC h2 coupling (Ge et al.) (colored by cytoarchitectural class); scatter of rsFC-rsFC heritability based on Solar versus rsFC – rsFC h2 coupling based on Ge et al. (2017) (Ge et al., 2017) (colored by cytoarchitectural class).

Results, P. 7

“and showed comparable patterns in an alternative measure of heritability of rsFC that accounts for intra-subject variations (Ge et al., 2017)”

In addition, we evaluated heritability of structure-function coupling, as suggested by the Reviewer. Heteromodal regions were less heritable than idiotypic, unimodal, and paralimbic regions. We now included this additional analysis in the *Results*, P.7.

“In addition, we assessed the heritability of MPC-rsFC coupling. Overall coupling was heritable ($h2=0.0260\pm0.088$), at the same time, we found that in particular coupling in heteromodal regions was less heritable relative to other cortical types (primary (idiotypic): $h2=0.284\pm0.073$; unimodal: 0.270 ± 0.093 ; heteromodal: 0.235 ± 0.087 ; paralimbic: 0.273 ± 0.082) with only differences between heteromodal and other cytoarchitectural class being below $p<0.05$ (idiotypic: $t=-4.158$, $p<0.001$; unimodal: $t=-3.223$, $p=0.006$; paralimbic: $t=-2.702$, $p=0.03$), (Fig 2).”

Fig 2. Genetic basis of structure-function coupling. C) Heritability of microstructure-function coupling, Right: distribution per cytoarchitectural class(M. Mesulam, 2000; Paquola, Vos De Wael, et al., 2019).

And *Discussion*, P.15.

“Our study, therefore, adds to a growing body of work suggest that coordinated coupling of structure and function itself is under genetic control and that this likely determines variation in cognitive phenotypes (Baum et al., 2020; Gu et al., 2021). Notably, studying heritability of coupling itself, we found heteromodal coupling to be least heritable, a pattern consistent with the spatial distribution of coupling heritability in previous work (Gu et al., 2021).”

Similar findings were observed using the approach of Ge et al., however we observed particularly low values in the ‘limbic’ functional network, possibly reflecting signal loss due to reduced scanner time per measurement.

Alternative heritability measurement of structure-function coupling

2) In the same vein, it is important to assess test-retest reliability of MPC, rsFC and the coupling measure. This can be done with the subset of ~40 individuals in the HCP study that had a second MRI scan 6 months after the first. This will also help disentangle the effect of measurement noise on the various quantities (see my previous comment).

As suggested, we tested the reliability of MPC, rsFC, and their coupling in the test-retest subset. We observed moderate test-retest reliability at an edge level (MPC ICC: $\text{mean} \pm \text{SD} = 0.430 \pm 0.059$; rsFC ICC: 0.549 ± 0.077), indicating that these metrics are stable over time, and these estimates are in line with previous reports (Noble, Scheinost, & Constable, 2019). Structure-function coupling itself was also moderately reliable (ICC 0.568 ± 0.221). At the same time, we found no relationship (Spearman’s $r = -0.02$) between coupling of MPC and rsFC and test-retest reliability, suggesting uncoupling of both measures does not relate to reduced reliability of either (Supplementary Fig. 15). We now included these Results in the *Supplementary Materials*, P.5.

“Evaluating test-retest reliability of MPC, rsFC and coupling of MPC and rsFC we observed overall moderate test-retest reliability across all measures (0.430 ± 0.059 ; rsFC ICC: 0.549 ± 0.077 ; coupling: 0.568 ± 0.221). At the same time, we found no relationship (Spearman’s $r = -0.02$) between coupling of MPC and rsFC and test-retest reliability, suggesting uncoupling of both measures does reflect reduced reliability of both (Supplementary Fig. 15).”

Supplementary Figure 15. Reliability of rsFC, MPC and rsFC-MPC coupling. A) *Left.* ICC of rsFC and association with node-level heritability of rsFC; *right.* ICC of MPC and association with node-level heritability of MPC; B) Reliability of structure-function coupling and association of strength of structure-function coupling with reliability of this measure.

And in *Methods*, P.28.

“To evaluate test-retest reliability of our measures we used the test-retest sample of HCP (Glasser et al., 2013) with complete T1wT2w and rsFC data at baseline and retest, leaving us with n=46, 32 females, age: mean±SD 30.196±3.364, range: 22-35, mean±SD days of test-retest interval: 139.304±68.994 days, range: 18-343 days. ICC was quantified using a Median Absolute Deviation Intraclass Correlation Coefficient; <https://warwick.ac.uk/fac/sci/statistics/staff/academic-research/nichols/scripts/matlab/madicc.m>.”

3) Apologies, but I am struggling to understand why assessing the correlation between the mean value of MPC/rsFC and their respective heritability estimates is helpful and how it can be interpreted (Figure 1D). Again, low coupling/MPC/rsFC being related to low heritability and high coupling/MPC/rsFC being related to high heritability may not indicate “genetic control” but could all be due to noise in the measures that impacts both quantities in the same direction. This needs to be controlled for (as suggested in my first comment) or at least assessed, possibly by showing an SNR map of the MRI measures from the test-retest analysis.

We thank the Reviewer for this remark. With regards to the analysis of heritability, we probed the correlation between mean value of MPC/rsFC and their respective heritability to understand whether regions that showed high correlation/covariance would also show high heritability, under the hypothesis that regions with high mean correlation/covariance would show more heritability than those with low values.

Further, we quantified the association between MPC/rsFC and test-retest reliability to evaluate whether mean-heritable regional correlation was similar to the association between mean and test-retest reliability of each measure (**Supplementary Figure 15**). Although there was variation in the relationship between mean and reliability for rsFC (mean±SD: 0.09±0.14), this pattern was not correlated with the association between mean and heritable FC (Spearman’s $r=0.07$, $p>0.1$). That is, row-wise associations between mean and ICC of rsFC did not correlate with row-wise associations of mean and heritability of rsFC. Performing the same analysis of MPC, we found high overall patterns of correspondence between mean and test-retest reliability of MPC (mean±SD: 0.66±0.09), indicating that regions with similar microstructural profiles overall showed high reliability whereas those with low MPC showed reduced reliability. We found no relationship between mean MPC – reliability of MPC and mean MPC-heritability of MPC (Spearman’s $r=0.18$, $p_{\text{spin}}>0.1$). We now report these analyses in the *Results*, P. 7.

“These associations were not significantly linked to test-retest reliability, affecting the measures in the same direction (Supplementary Results).”

And *Supplementary Results*, P. 5:

“To control for the possibility that the association between mean and heritability of the measurements was related to an association with noise, measured by means of test-retest reliability in the retest sample of the HCP dataset, going in the same direction, we computed the parcel-wise relationship between each measure and their respective reliability, as quantified by ICC. Here we observed that while there was variation across parcels in the relationship between mean and reliability for rsFC (mean±SD: 0.09±0.14), this pattern was not correlated with the association between mean and heritable FC (Spearman’s $r=0.07$, $p>0.1$). Performing the same analysis of MPC, we found correspondence between mean MPC and the ICC of MPC (mean±SD: 0.66±0.09), indicating that regions with similar microstructural profiles overall showed high test-retest reliability. Again, we found no

relationship between mean MPC – reliability of MPC and mean MPC-heritability of MPC (Spearman's $r=0.18$, $p_{spin}>0.1$).”

And *Methods*, P. 27.

“To evaluate test-retest reliability of our measures we used the test-retest sample of HCP (Glasser et al., 2013) with complete T1wT2w and rsFC data at baseline and retest, leaving us with $n=46$, 32 females, age: mean/SD 30.196/3.364, range: 22-35, mean days of test-retest interval: 139.304 days, SD 68.994, min-max: 18-343 days. ICC was quantified using a Median Absolute Deviation Intraclass Correlation Coefficient; <https://warwick.ac.uk/fac/sci/statistics/staff/academic-research/nichols/scripts/matlab/madicc.m>.”

Supplementary Fig 15. Correspondence of test-retest ICC with rsFC and MPC. C). Seed-wise correlation between mean rsFC and ICC of rsFC; D). Seed-wise correlation between mean rsFC and ICC of MPC.

4) It is unclear how the node-wise measures in Figure 1D are being calculated.

We are happy to further clarify, we computed heritability of the region-to-region rsFC/MPC values *i.e.*, for each node in the MPC/rsFC matrix. Following with used this node-based measure for our further analysis. We have now further clarified this in the revised *Results*, P. 7.

“We first computed the node-wise heritability of MPC and rsFC using Sequential Oligogenic Linkage Analysis Routines (www.solar-eclipse-genetics.org; Solar Eclipse 8.4.0).”

5) Figure 1E is also a bit confusing to me, I am struggling to understand why the correlation between coupling and the correlation between the mean and heritability of MPC/rsFC is being assessed. It seems more interpretable and straightforward to instead look at the correlation between structure-function coupling and the two heritability estimates of MPC and rsFC. Then, it would support the current claim in the paper:

“In other words, reduced structure-function coupling in human transmodal regions was paralleled by reductions in genetic control over both MPC and rsFC.”

My confusion could also be a result of my lack of understanding of the motivation to correlate mean and heritability values (see comment #3). It seems this correlation is being used to assess “genetic control” but isn’t that what heritability is providing? At minimum, please better motivate this part of the analysis and explain the reasoning in a more detailed manner.

We are happy to further clarify. The main aim of this step was to evaluate potential variability in association of heritability of connectivity profiles and mean of connectivity profiles from

idiotypic to paralimbic regions in either or both measures. In other words, we assessed associations between mean connectivity profiles and heritability of profiles in a row-wise fashion, and stratified findings by cytoarchitectural class. We, thus, aimed to quantify to what extent genetic factors shape the connectivity profiles of the respective measures. This would provide us with a local index of ‘genetic control’ of a given seed from a network perspective. We have now also computed heritability of coupling itself, providing an alternative approach to evaluate genetic control of structure- function coupling (please see R3Q1). We further explained this in the *Results*, P. 7.

“To assess to what extent rsFC and MPC interregional patterns were under genetic control, we compared mean seed-wise connectivity/covariance profiles with seed-wise heritability of each measure separately. This index provides us with a local measure to what extent mean and heritable patterns are similar, analogous to the MPC-rsFC coupling measure.”

In addition, we now tested for differences in coupling between microstructural classes. We found that for genetic coupling in MPC, only heteromodal regions showed consistently reduced (FDR_q<0.05) coupling relative to other classes (idiotypic: $t=-12.363$, $p<0.001$; unimodal: $t=-6.129$, $p=0.006$; paralimbic: $t=-4.022$, $p=0.03$). In case of rsFC we found both transmodal classes to show reductions in genetic coupling relative to non-transmodal classes (heteromodal: idiotypic: $t=-9.320$, $p<0.001$; unimodal: $t=-6.062$, $p<0.001$ and paralimbic regions (idiotypic: $t=-6.340$, $p<0.001$; unimodal: $t=-2.780$, $p=0.024$). Genetic coupling between heteromodal and paralimbic regions showed no difference in rsFC ($t=1.839$, $p_{\text{uncorr}}=0.067$). We have added this information on *Results*, P. 7.

“Assessing differences in genetic coupling as a function of cytoarchitectural class, we found that for MPC, only heteromodal regions showed consistently reduced (FDR_q<0.05) coupling relative to other classes (idiotypic: $t=-12.363$, $p<0.001$; unimodal: $t=-6.129$, $p=0.006$; paralimbic: $t=-4.022$, $p=0.03$). For rsFC transmodal classes displayed reductions in genetic coupling relative to non-transmodal classes (heteromodal: idiotypic: $t=-9.320$, $p<0.001$; unimodal: $t=-6.062$, $p<0.001$ and paralimbic regions (idiotypic: $t=-6.340$, $p<0.001$; unimodal: $t=-2.780$, $p=0.024$). Genetic coupling between heteromodal and paralimbic regions showed no significant difference in rsFC ($t=1.839$, $p_{\text{uncorr}}=0.067$).”

And *Methods*, P. 21.

“A similar approach quantified genetic coupling, i.e. we correlated row-wise mean MPC/rsFC with the row-wise heritability maps.”

6) *The analysis with the macaques is an important one, but there are key differences in data acquisition that may be influencing the results. It is known that anesthesia can cause tighter coupling of structural connectivity and functional connectivity (Demertzi, 2019). At minimum, this should be considered in the discussion or by adding a limitations paragraph (which is currently not included, but would certainly help the reader fully understand the implications of the study).*

Demertzi A, Tagliazucchi E, Dehaene S, et al. Human consciousness is supported by dynamic complex patterns of brain signal coordination. Sci Adv. 2019;5(2):eaat7603. doi:10.1126/sciadv.aat7603

Of particular note regarding this issue, in Figure 4A the higher-order fronto-parietal and DMN regions seem to have stronger structure-function coupling in macaques (upper left quadrant in the macaque panel). Further, the left-hand side of the quadrant that is dominated by rsFC seems to have stronger coupling in macaques. This could be due to the anesthesia causing a collapse of the rsFC to the anatomical networks, particularly for these higher-order regions that likely underlie consciousness.

We thank the Reviewer for this suggestion. Our main analysis utilized rsFC data from anesthetized macaques. However, when additionally assessing other macaque samples with different age and sex distributions (Davis: anesthetized, N=19, all female, age=20.38±0.93 yrs; Newcastle: awake, N=10, 8 males, age=8.28±2.33 yrs; Oxford: anesthetized, N=19, all male, age=4.01±0.98 yrs). Despite some inter-site differences, patterns of decoupling were comparable for anesthetized and awake samples (Davis-Newcastle: $r=0.78$; Davis-Oxford: $r=0.90$; Newcastle-Oxford: $r=0.71$). All samples showed correlation with the human HCP measure of structure-function coupling (Davis: Spearman's $r=0.48$, $p_{\text{spin}}=0.016$; Newcastle: Spearman's $r=0.36$, $p_{\text{spin}}=0.06$; Oxford: Spearman's $r=0.40$, $p_{\text{spin}}=0.05$). Similar to our main findings, we observed high coupling for all macaque samples in idiosyncratic, unimodal, and heteromodal regions, whereas only paralimbic regions were uncoupled. Conversely, in human data we observed relative uncoupling in unimodal, heteromodal, and paralimbic regions.

Sample	Idiosyncratic [mean(SD)]	Unimodal	Heteromodal	Paralimbic
HCP	0.465(0.181)	0.149(0.263)	0.063(0.198)	0.012(0.146)
Davis	0.376(0.260)	0.346(0.221)	0.333(0.255)	0.052(0.230)
Newcastle	0.322(0.167)	0.262(0.156)	0.249(0.222)	0.108(0.245)
Oxford	0.433(0.255)	0.390(0.224)	0.403(0.269)	0.166(0.284)

Supplementary Table 2. Structure-function coupling mean(SD) as a function of cytoarchitectural class in humans and macaques.

Sample	Idiosyncratic [mean(SD)]	Unimodal	Heteromodal	Paralimbic
Davis- HCP	0.0628 ($p>0.1$)	6.399 ($p<0.001$)	6.445 ($p<0.001$)	0.063 ($p>0.1$)
Newcastle - HCP	-0.6713 ($p>0.1$)	4.516 ($p<0.001$)	4.759 ($p<0.001$)	2.184 ($p>0.1$)
Oxford - HCP	0.699 ($p>0.1$)	7.744 ($p<0.001$)	7.905 ($p<0.001$)	3.157 ($p=0.01$)

Supplementary Table 3. Difference in structure-function coupling between humans and macaque across samples

Supplementary Fig 7. Structure-function coupling in different macaque samples. A) Anesthetized macaque (sample: Davis) structure-function coupling and correlation with human structure-function coupling projected to macaque space; **B)** Awake macaque (sample Newcastle) structure-function coupling and correlation with human structure-function coupling projected to macaque space; **C)** Anesthetized macaque (sample Oxford) structure-function coupling and correlation with human structure-function coupling projected to macaque space; **D)** Structure-function coupling averaged in cytoarchitectural classes (M. Mesulam, 2000).

Results, P. 9:

“Replacing the macaque rsFC matrix with two other samples, including males and females with a different age-range, and awake as well as anesthetized monkeys, yielded broadly similar patterns of structure-function coupling (Supplementary Results).”

Methods P. 23:

“To evaluate robustness of the macaque findings, we evaluated two additional macaque samples with rsFC data openly available in PRIME-DE (http://fcon_1000.projects.nitrc.org/indi/indiPRIME.html) (Milham et al., 2018), previously used in (T. Xu et al., 2020). The samples varied in age-range, sex, and acquisition parameters. The Newcastle dataset (awake (Rinne et al., 2017; Schonwiesner et al., 2015)) consisted of 10 rhesus macaques (8 males, age mean \pm SD=8.28 \pm 2.33, weight=11.76 \pm 3.38) scanned on a Vertical Bruker 4.7T primate scanner. The fMRI session was acquired with 1.2 \times 1.2 \times 1.2 mm resolution, TR=2,000 ms, 8.33-min per scan (250 volumes \times 2 scan) per animal. No contrast-agent was used during the scans. In the case of the Oxford dataset (anesthetized), we included nineteen rhesus macaques with preprocessing and surface reconstruction as in previous work (T. Xu et al., 2020)(all males, age=4.01 \pm 0.98 years, weight=6.61 \pm 2.04 kg). The macaques were scanned on a 3T with a 4-channel coil (Noonan et al., 2014). Resting-state fMRI (rs-fMRI) data were collected with 2 mm isotropic resolution, TR=2,000 ms, 53.3 min (1,600 volumes). No contrast-agent was used during the scans.”

Supplementary Results, P. 3:

“To evaluate whether the observed structure-function coupling in macaques might be due to awake versus anesthetized status, as well as differences in age-range and sex between samples, we evaluated two additional macaque samples, the Newcastle and Oxford sample (**Supplementary Fig. 7**). As these datasets did not have high-resolution microstructural MRI data available, we correlated the region-wise average functional connectivity data of the two additional samples, thought to be most impacted by anesthesia, with the microstructural profile covariance data of the Davis sample. Despite some inter-site differences, patterns of decoupling were similar for anesthetized and awake samples with different sex and age distributions (Davis-Newcastle: $r=0.78$; Davis-Oxford: $r=0.90$; Newcastle-Oxford: $r=0.71$). All samples showed correlation with the human HCP measure of structure-function coupling (Davis: Spearman’s $r=0.48$, $p_{spin}=0.012$; Newcastle: Spearman’s $r=0.36$, $p_{spin}=0.06$; Oxford: Spearman’s $r=0.40$, $p_{spin}=0.05$). For all macaque samples, we observed similarly high coupling in idiosyncratic, unimodal and heteromodal regions, whereas only paralimbic regions were uncoupled. Conversely, in human data we observed relative uncoupling in unimodal, heteromodal and paralimbic regions, (**Supplementary Table 2 and 3**).”

At the same time, we agree that acknowledging limitations benefits the manuscript, particularly in regards to potential effect of anesthesia on structure-function coupling, *Discussion*, P.17.

“Moreover, previous work has indicated differences in brain function at rest relate to the level of consciousness in humans and macaques (Demertzi et al., 2019; T. Xu et al., 2019), and also our findings suggest subtle differences in coupling of structure and function between awake and anesthetized macaques. Despite this variation, we observed a consistent difference in unimodal and heteromodal coupling between (awake) humans and awake as well as anesthetized macaques. Future work may expand on this work to further explore variations in structure-function coupling by level of consciousness, particularly in humans by exploiting the capacity for introspection. Such analysis may provide additional insight in the relation between structure-function coupling and features of internal cognition.”

7) Related to the above comment, it appears that all female macaques were used in the non-human primate data. Are there sex differences in coupling in the human data? If there do exist sex differences in structure-function coupling (in Gu, et al (2021), regionally-varying sex differences in structure-function coupling were in fact identified) then the comparison of human to non-human data could be biased. At minimum, an assessment of sex differences in coupling in humans should be provided and limitations in comparing these values to an all-female macaque dataset should be discussed.

We thank the Reviewer for this remark. To evaluate whether potential differences observed between humans and macaques could be due to sex differences between both samples, we

studied an additional macaque samples with variable sex distributions (Davis: anesthetized, N=19, all female, age=20.38±0.93 yrs; Newcastle: awake, N=10, 8 males, age=8.28±2.33 yrs; Oxford: anesthetized, N=19, all male, age=4.01±0.98 yrs) (**Supplementary Fig 7**). Indeed, patterns of decoupling were similar for anesthetized and awake samples with different sex and age distributions (Davis-Newcastle: $r=0.78$; Davis-Oxford: $r=0.90$; Newcastle-Oxford: $r=0.71$). All samples showed correlation with the human HCP measure of structure-function coupling, mapped to macaque space (Davis: Spearman's $r=0.48$, $p_{\text{spin}}=0.012$; Newcastle: Spearman's $r=0.36$, $p_{\text{spin}}=0.06$; Oxford: Spearman's $r=0.40$, $p_{\text{spin}}=0.05$). The consistency of patterns across macaque samples suggests that differences observed between humans and macaques could not be explained by sex differences. We reported this analysis in the *Supplementary Results*, P. 3.

*“To evaluate whether the observed difference in structure-function coupling between humans and macaques might be due to awake versus anesthetized status, as well as differences in age-range and sex between samples, we evaluated two additional macaque samples, the Newcastle and Oxford sample (**Supplementary Fig 7**). As these datasets did not have high-resolution microstructural MRI data available, we correlated the region-wise average functional connectivity data of the two additional samples, thought to be most impacted by anesthesia, with the microstructural profile covariance data of the Davis sample. Despite some inter-site differences, patterns of decoupling were similar for anesthetized and awake samples with different sex and age distributions (Davis-Newcastle: $r=0.78$; Davis-Oxford: $r=0.90$; Newcastle-Oxford: $r=0.71$). All samples showed correlation with the human HCP measure of structure-function coupling (Davis: Spearman's $r=0.48$, $p_{\text{spin}}=0.012$; Newcastle: Spearman's $r=0.36$, $p_{\text{spin}}=0.06$; Oxford: Spearman's $r=0.40$, $p_{\text{spin}}=0.05$). For all macaque samples, we observed similarly high coupling in idiosyncratic, unimodal and heteromodal regions, whereas only paralimbic regions were uncoupled. Conversely, in human data we observed relative uncoupling in unimodal, heteromodal and paralimbic regions (**Supplementary Table 2 and 3**).”*

Studying the difference in structure-function coupling, we found that in both males and females there was a highly comparable uncoupling of rsFC and MPC ($r=0.99$). However, directly comparing both sexes, we also observed some differences. We have now included these findings in the *Supplementary Results*, P.4.

*“To study whether observed patterns of structure-function coupling in humans were similar between males and females we evaluate coupling patterns in both sexes (**Supplementary Fig 11**). We observed highly similar patterns of coupling and uncoupling ($r=0.99$). At the same time, we found also differences as function of sex at the individual level, notably increased coupling in superior frontal and parietal regions and decreased coupling in temporal, cingulate and medial frontal regions in males relative to females ($FDRq<0.05$).”*

And *Discussion*, P.17.

“Secondly, structure-function coupling shows some association with individual variations such as sex and behavioral traits and states. Indeed, though uncoupling was similar in both sexes in humans and macaques, ours and previous work also report subtle differences in coupling between (human) males and females. Such patterns may reflect sex-biased brain development (S. Liu, Seidlitz, Blumenthal, Clasen, & Raznahan, 2020) and hormonal variations (Taylor et al., 2020).”

Supplementary Fig 11. Sex differences in structure-function coupling. A) Structure-function coupling in females and B) males, and *right* their intercorrelation; C) Difference between males and females, red indicates males have stronger coupling than females and blue indicates females have stronger coupling than males. Black outline indicates differences at $FDRq < 0.05$.

8) *Figure 3A could again be driven by the varying levels of noise, please see my previous comments for ways to try and account for or assess the influence of noise in both the MRI-based measures and the heritability measures. Please also clarify what is being shown in Figure 3A in the heatmaps in the bottom row.*

We now computed test-retest reliability of the rsFC and MPC principal gradients in humans and their difference. Gradients and their difference had good test-retest reliability (MPC G_1 ICC $mean \pm SD$: 0.71 ± 0.21 ; rsFC G_1 : 0.60 ± 0.23 ; MPC $G_1 - rsFC G_1$: 0.66 ± 0.20). Moreover, we did not observed a relationship between reliability and gradient scores (MPC G_1 $r = 0.20$, $p_{spin} > 0.1$; rsFC G_1 $r = 0.19$, $p_{spin} > 0.1$; MPC $G_1 - rsFC G_1$ $r = 0.00$), suggesting only minimal relationships with noise on both measures and their differential organization. We included this test-retest analysis in the revised *Results*, P. 10;

“Organizational gradients did not correlate with test-retest intraclass correlations of the respective measures (Supplementary Results) suggesting that differences are likely not linked to variable levels of noise.”

And *Supplementary Results*, P. 5:

“To assess whether both gradients and their difference also reflected variation in noise, we evaluated the test-retest reliability of each gradient and their difference using ICC. Overall, gradients were reliable (MPC G_1 ICC $mean \pm std$: 0.71 ± 0.21 ; rsFC G_1 : 0.60 ± 0.23 ; MPC $G_1 - rsFCG_1$: 0.66 ± 0.20) and we observed no relationship between reliability and gradient loadings (MPC G_1 $r = 0.21$, $p_{spin} > 0.1$; rsFC G_1 $r = 0.19$, $p_{spin} > 0.1$; MPC $G_1 - rsFCG_1$ $r = 0.00$).”

Supplementary Fig 16. Intra-class correlation coefficient (ICC) of MPC and rsFC principal gradients and their difference. A) ICC of MPC-G1; B) ICC of rsFC0G1; C). ICC of MPC-G1 and rsFC-G1 difference.

The bottom row at Figure 3A displays average heritability values between the 10 respective gradient bins for each measure. Along the gradient, the highest heritability is observed between regions at similar gradient loadings (using 10 equally sized bins along the respective MPC/rsFC gradient). We have now clarified this in the *Supplementary Results* as we moved this part of the Figure to the Supplement based on the recommendations of Reviewer 2.

“We evaluated whether heritability of rsFC and MPC showed similar patterning as gradients over the mean of each respective measure. Overall, we found highly similar patterns. Principal axes of mean and heritable highly correlated for both MPC ($r=0.808$, $p_{spin}<0.0001$) and for rsFC ($r=0.892$, $p_{spin}<0.0001$) and regions at similar levels of the gradient showed also heightened heritability of MPC/rsFC with regions places at similar gradient levels.”

9) Are there any covariates that should be considered in the rsFC, MPC or coupling analysis? In-scanner motion, age, sex, cognitive scores? In the discussion there is some mention of structure-function decoupling being linked to cognition as determined by the NeuroSynth analysis; there are cognitive scores for these subjects and this hypothesis could also be tested over the population of humans in the HCP study.

Our main findings focused on group-level patterns in humans, in combination with heritability analyses and cross-species comparisons to assess uncoupling of structure and function in the transmodal cortex. However, structure-function coupling is similar at the level of the individual (Supplementary Fig 10), similar in females and males (Supplementary Figure 11) and varies only little as a function of ICV (Supplementary Figure 10). While we performed a meta-analytical decoding to further qualify how the group-level maps relate to distinct functional processes (Margulies et al., 2016; Paquola, Vos De Wael, et al., 2019), this approach is different from an examination of which inter-individual differences in structure-function coupling relate to inter-individual differences in behavior (Genon, Bernhardt, La Joie, Amunts, & Eickhoff, 2021). Such an approach would not embed task-based activations within our 2D coupling space, but rather investigate correlations between variations in

respective coupling markers and in turn their link to the coupling measure and difference between rsFC and MPC principal gradients.

To also evaluate the association with inter-individual differences in behavior and maps of structure-function coupling and gradient differences between MPC and rsFC, we correlated relevant behavioral scores of 20 behavioral markers with individual maps of coupling and MPC-rsFC G1 difference (total Cognition, card sorting, list sorting, friendship, picture vocabulary, reading English, pain, endurance, flanker, picture sequence, self-efficiency, perceived stress, noise, SCPT, sadness, and NEO-FFI (Van Essen et al., 2013)). Next, we explored whether their cortex-wide t-maps could be associated with the coupling and gradient difference patterns. We found that self-efficiency, friendship and extraversion related to strong uncoupling, e.g. more coupling in sensory regions and uncoupling in transmodal regions. Along the axes dissociating heteromodal from paralimbic transmodal regions, we found a higher similarity (decreased differences) of both associated with stress, sadness, pain and neuroticism. In turn, increased differences of both were related to working memory, cognition and card sorting. Though these patterns should be interpreted with care, they are again suggestive of meaningful dimensions related to higher-order human cognition. We have now included these additional findings in the *Supplementary Materials*, P. 4.

“To evaluate whether coupling and gradient differences in structure and function were meaningful at the individual level for behavior, we correlated relevant behavioral scores of the HCP dataset with individual maps of coupling and MPC-rsFC G1 difference and assessed whether t-values correlated with the coupling and gradient difference pattern. We found that self-efficiency, friendship and extraversion related to strong uncoupling, e.g. more coupling in sensory regions and uncoupling in transmodal regions. Along the axes dissociating heteromodal from paralimbic transmodal regions, we found a higher similarity (decreased differences) of MPC G1 and rsFC G1 could be associated with stress, sadness, pain and neuroticism. In turn, increased difference between both gradients was related to working memory, cognition and card sorting. These patterns should be interpreted differently from the mapping with meta-analytical task activations. However, they are again suggestive of meaningful dimensions in structure-function coupling and transmodal uncoupling related to higher-order human cognition (Supplementary Fig 12).”

Results, P. 13.

“Finally, we explored how individual differences in behavior, measured using task batteries included in HCP, were differentiated along the two axes identified in our study. In particular, decreased differentiation between principal gradients of MPC and rsFC related to “stress” and “pain”, whereas increased difference related to “working memory” and “cognition”. This observation suggests that the dimensions identified by our analysis may have behavioral relevance, an important hypothesis for future studies to examine (Supplementary Results).”

Methods, P. 26.

“To assess the association between structure function coupling and gradient difference and individual-level behavior we selected 20 markers of individual difference from the HCP battery; total Cognition, card sorting, list sorting, friendship, picture vocabulary, reading English, pain, endurance, flanker, picture sequence, self-efficiency, perceived stress, noise, SCPT, sadness, and NEO-FFI (Van Essen et al., 2013). Subsequently, we performed linear regression using SurfStat (Worsley et al., 2009) to probe association between parcel-level variation in coupling and gradient difference, while controlling for age and sex. Then we correlated the 20-task-based t-maps with mean maps of coupling difference and gradient differences and visualized them in 2D space to evaluate their relationship to patterns of coupling and gradient difference.”

Supplementary Fig 12. Individual level correlations with structure-function coupling along 2D framework.

Minor comments:

1) What type of correlation was used to calculate the structure-function coupling measure (Pearson or Spearman) and was this choice justified by the shapes of the distributions? It states that in the heritability analysis, the FC and MPC values were transformed to adhere to normality assumptions; were the transformed values also used in the coupling calculation (if Pearson’s correlation that assumes normality was used)?

For our analyses of structure-function coupling we used Spearman’s rank correlation, given that we aimed to test the relationship between MPC and rsFC without distribution or scaling assumptions making rank-correlation more robust. Nevertheless, computing the association using product moment (*i.e.*, Pearson) correlations gave highly similar results in humans ($r=0.95$) and macaques ($r=0.88$).

2) Particularly in genetic analysis, one needs to determine if race is a driving factor. Race could be included as a co-variate in the heritability analysis or the heritability analysis should be reproduced using a homogenous group of Caucasian individuals.

We reran the heritability analysis in individuals whose self-reported race was ‘White’ (n=749 individuals in our main sample). Similar patterns of ‘genetic uncoupling’ were observed as in

the complete sample $rsFC-rsFC_{\text{heritability}}: r=0.95$ and $MPC-MPC_{\text{heritability}}: r=0.93$, $rsFC_{G1} r: 0.975$ and $MPC_{G1} r: 0.935$, suggesting observed patterns were consistent between Caucasian subsample and the complete sample. We have added this information to the *Supplementary Results*, P.4.

“To evaluate the potential impact of ethnicity on heritability, we reran the heritability analysis in individuals whose self-reported race was ‘White’ (n=749 individuals in our main sample). Indeed, similar patterns of ‘genetic uncoupling’ were observed as in the complete sample $rsFC-rsFC_{\text{heritability}}: r=0.95$ and $MPC-MPC_{\text{heritability}}: r=0.93$, $rsFC_{G1} r: 0.975$ and $MPC_{G1} r: 0.935$ (Supplementary Fig 13).”

Supplementary Fig 13. Heritability observations in Caucasian subsample. A) Correspondence between mean and heritable rsFC in Caucasian subsample and similarity with full study sample; B) Correspondence between mean and heritable MPC in Caucasian subsample and similarity with full study sample; C) Gradients of rsFC and MPC in Caucasian subsample and similarity between heritable gradients and those of heritable gradients of full study sample for the respective measure.

3) In the transcriptomic analysis, it is stated that genes showing consistent whole-brain expression patterns across donors ($r>0.5$) were retained; however, there is quite a sex imbalance in the Allen brain atlas data that could result in a bias toward only investigating genes that have higher expression rates in males. At a minimum, this limitation should be discussed.

We agree and now explicitly acknowledge potential limitations of the AHBA in the *Discussion* (P.17):

“It is of note that although the Allen human brain atlas is the most densely sampled transcriptomic dataset of the human brain currently available, it comes with various limitations, such as sex and age imbalance (Arnatkeviciute, Fulcher, & Fornito, 2019; Hawrylycz et al., 2012). Thus, findings from our study relating to gene expression should be interpreted with caution. In particular, although we selected only genes that were

relatively consistent across donors, there may still be a bias toward those genes having higher expression rates in males than in females.”

4) The fourth paragraph of text in the Supplemental material section seems to be out of place?

We thank the Reviewer for the suggestion, and removed the paragraph.

For transparency, I will de-anonymize my review.

-Amy Kuceyeski

We thank Dr Kuceyeski for the positive evaluation and helpful comments.

REFERENCES FOR RESPONSES

- Arnatkeviciute, A., Fulcher, B. D., & Fornito, A. (2019). A practical guide to linking brain-wide gene expression and neuroimaging data. *Neuroimage*, *189*, 353-367. doi:10.1016/j.neuroimage.2019.01.011
- Assem, M., Glasser, M. F., Van Essen, D. C., & Duncan, J. (2020). A Domain-General Cognitive Core Defined in Multimodally Parcellated Human Cortex. *Cereb Cortex*, *30*(8), 4361-4380. doi:10.1093/cercor/bhaa023
- Barks, S. K., Parr, L. A., & Rilling, J. K. (2015). The default mode network in chimpanzees (Pan troglodytes) is similar to that of humans. *Cereb Cortex*, *25*(2), 538-544. doi:10.1093/cercor/bht253
- Barton, R. A., & Venditti, C. (2013). Human frontal lobes are not relatively large. *Proc Natl Acad Sci U S A*, *110*(22), 9001-9006. doi:10.1073/pnas.1215723110
- Baum, G. L., Cui, Z., Roalf, D. R., Ciric, R., Betzel, R. F., Larsen, B., . . . Satterthwaite, T. D. (2020). Development of structure-function coupling in human brain networks during youth. *Proc Natl Acad Sci U S A*, *117*(1), 771-778. doi:10.1073/pnas.1912034117
- Buckner, R. L., & Krienen, F. M. (2013). The evolution of distributed association networks in the human brain. *Trends Cogn Sci*, *17*(12), 648-665. doi:10.1016/j.tics.2013.09.017
- Burt, J. B., Demirtas, M., Eckner, W. J., Navejar, N. M., Ji, J. L., Martin, W. J., . . . Murray, J. D. (2018). Hierarchy of transcriptomic specialization across human cortex captured by structural neuroimaging topography. *Nat Neurosci*, *21*(9), 1251-1259. doi:10.1038/s41593-018-0195-0
- Cox, R. W. (1996). AFNI: software for analysis and visualization of functional magnetic resonance neuroimages. *Comput Biomed Res*, *29*(3), 162-173. doi:10.1006/cbmr.1996.0014
- Dart, R. A. (1934). The Dual Structure of the Neopallium: its History and Significance. *J Anat*, *69*(Pt 1), 3-19. Retrieved from <https://www.ncbi.nlm.nih.gov/pubmed/17104513>
- Demertzi, A., Tagliazucchi, E., Dehaene, S., Deco, G., Barttfeld, P., Raimondo, F., . . . Sitt, J. D. (2019). Human consciousness is supported by dynamic complex patterns of brain signal coordination. *Sci Adv*, *5*(2), eaat7603. doi:10.1126/sciadv.aat7603
- Demirtas, M., Burt, J. B., Helmer, M., Ji, J. L., Adkinson, B. D., Glasser, M. F., . . . Murray, J. D. (2019). Hierarchical Heterogeneity across Human Cortex Shapes Large-Scale Neural Dynamics. *Neuron*, *101*(6), 1181-1194 e1113. doi:10.1016/j.neuron.2019.01.017
- Donahue, C. J., Glasser, M. F., Preuss, T. M., Rilling, J. K., & Van Essen, D. C. (2018). Quantitative assessment of prefrontal cortex in humans relative to nonhuman primates. *Proc Natl Acad Sci U S A*, *115*(22), E5183-E5192. doi:10.1073/pnas.1721653115
- Eichert, N., Robinson, E. C., Bryant, K. L., Jbabdi, S., Jenkinson, M., Li, L., . . . Mars, R. B. (2020). Cross-species cortical alignment identifies different types of anatomical reorganization in the primate temporal lobe. *elife*, *9*. doi:10.7554/eLife.53232
- Ganzetti, M., Wenderoth, N., & Mantini, D. (2014). Whole brain myelin mapping using T1- and T2-weighted MR imaging data. *Front Hum Neurosci*, *8*, 671. doi:10.3389/fnhum.2014.00671
- Ge, T., Holmes, A. J., Buckner, R. L., Smoller, J. W., & Sabuncu, M. R. (2017). Heritability analysis with repeat measurements and its application to resting-state functional connectivity. *Proc Natl Acad Sci U S A*, *114*(21), 5521-5526. doi:10.1073/pnas.1700765114
- Genon, S., Bernhardt, B. C., La Joie, R., Amunts, K., & Eickhoff, S. B. (2021). The many dimensions of human hippocampal organization and (dys)function. *Trends Neurosci*, *44*(12), 977-989. doi:10.1016/j.tins.2021.10.003

- Glasser, M. F., Coalson, T. S., Robinson, E. C., Hacker, C. D., Harwell, J., Yacoub, E., . . . Van Essen, D. C. (2016). A multi-modal parcellation of human cerebral cortex. *Nature*, *536*(7615), 171-178. doi:10.1038/nature18933
- Glasser, M. F., Sotiropoulos, S. N., Wilson, J. A., Coalson, T. S., Fischl, B., Andersson, J. L., . . . Consortium, W. U.-M. H. (2013). The minimal preprocessing pipelines for the Human Connectome Project. *Neuroimage*, *80*, 105-124. doi:10.1016/j.neuroimage.2013.04.127
- Glasser, M. F., & Van Essen, D. C. (2011). Mapping human cortical areas in vivo based on myelin content as revealed by T1- and T2-weighted MRI. *J Neurosci*, *31*(32), 11597-11616. doi:10.1523/JNEUROSCI.2180-11.2011
- Greve, D. N., & Fischl, B. (2009). Accurate and robust brain image alignment using boundary-based registration. *Neuroimage*, *48*(1), 63-72. doi:10.1016/j.neuroimage.2009.06.060
- Gu, Z., Jamison, K. W., Sabuncu, M. R., & Kuceyeski, A. (2021). Heritability and interindividual variability of regional structure-function coupling. *Nat Commun*, *12*(1), 4894. doi:10.1038/s41467-021-25184-4
- Hawrylycz, M. J., Lein, E. S., Guillozet-Bongaarts, A. L., Shen, E. H., Ng, L., Miller, J. A., . . . Jones, A. R. (2012). An anatomically comprehensive atlas of the adult human brain transcriptome. *Nature*, *489*(7416), 391-399. doi:10.1038/nature11405
- Hayashi, T., Hou, Y., Glasser, M. F., Autio, J. A., Knoblauch, K., Inoue-Murayama, M., . . . Van Essen, D. C. (2021). The nonhuman primate neuroimaging and neuroanatomy project. *Neuroimage*, *229*, 117726. doi:10.1016/j.neuroimage.2021.117726
- Heuer, K., Gulban, O. F., Bazin, P. L., Osoianu, A., Valabregue, R., Santin, M., . . . Toro, R. (2019). Evolution of neocortical folding: A phylogenetic comparative analysis of MRI from 34 primate species. *Cortex*, *118*, 275-291. doi:10.1016/j.cortex.2019.04.011
- Jenkinson, M., Beckmann, C. F., Behrens, T. E., Woolrich, M. W., & Smith, S. M. (2012). Fsl. *Neuroimage*, *62*(2), 782-790. doi:10.1016/j.neuroimage.2011.09.015
- Liu, C., Yen, C. C., Szczupak, D., Ye, F. Q., Leopold, D. A., & Silva, A. C. (2019). Anatomical and functional investigation of the marmoset default mode network. *Nat Commun*, *10*(1), 1975. doi:10.1038/s41467-019-09813-7
- Liu, S., Seidlitz, J., Blumenthal, J. D., Clasen, L. S., & Raznahan, A. (2020). Integrative structural, functional, and transcriptomic analyses of sex-biased brain organization in humans. *Proc Natl Acad Sci U S A*, *117*(31), 18788-18798. doi:10.1073/pnas.1919091117
- Lorio, S., Kherif, F., Ruef, A., Melie-Garcia, L., Frackowiak, R., Ashburner, J., . . . Draganski, B. (2016). Neurobiological origin of spurious brain morphological changes: A quantitative MRI study. *Hum Brain Mapp*, *37*(5), 1801-1815. doi:10.1002/hbm.23137
- Marcus, D. S., Harwell, J., Olsen, T., Hodge, M., Glasser, M. F., Prior, F., . . . Van Essen, D. C. (2011). Informatics and data mining tools and strategies for the human connectome project. *Front Neuroinform*, *5*, 4. doi:10.3389/fninf.2011.00004
- Margulies, D. S., Ghosh, S. S., Goulas, A., Falkiewicz, M., Huntenburg, J. M., Langs, G., . . . Smallwood, J. (2016). Situating the default-mode network along a principal gradient of macroscale cortical organization. *Proc Natl Acad Sci U S A*, *113*(44), 12574-12579. doi:10.1073/pnas.1608282113
- Mesulam, M. (2000). Behavioral neuroanatomy: Largescale networks, association cortex, frontal syndromes, the limbic system, and hemispheric specialization. In *In: Principles of Behavioral and Cognitive Neurology* (pp. 1-120).
- Mesulam, M. M. (1998). From sensation to cognition. *Brain*, *121* (Pt 6), 1013-1052. doi:10.1093/brain/121.6.1013

- Milham, M. P., Ai, L., Koo, B., Xu, T., Amiez, C., Balezeau, F., . . . Schroeder, C. E. (2018). An Open Resource for Non-human Primate Imaging. *Neuron*, *100*(1), 61-74 e62. doi:10.1016/j.neuron.2018.08.039
- Murray, J. D., Bernacchia, A., Freedman, D. J., Romo, R., Wallis, J. D., Cai, X., . . . Wang, X. J. (2014). A hierarchy of intrinsic timescales across primate cortex. *Nat Neurosci*, *17*(12), 1661-1663. doi:10.1038/nn.3862
- Nenning, K. H., Xu, T., Schwartz, E., Arroyo, J., Woehrer, A., Franco, A. R., . . . Langs, G. (2020). Joint embedding: A scalable alignment to compare individuals in a connectivity space. *Neuroimage*, *222*, 117232. doi:10.1016/j.neuroimage.2020.117232
- Noble, S., Scheinost, D., & Constable, R. T. (2019). A decade of test-retest reliability of functional connectivity: A systematic review and meta-analysis. *Neuroimage*, *203*, 116157. doi:10.1016/j.neuroimage.2019.116157
- Noonan, M. P., Sallet, J., Mars, R. B., Neubert, F. X., O'Reilly, J. X., Andersson, J. L., . . . Rushworth, M. F. (2014). A neural circuit covarying with social hierarchy in macaques. *PLoS Biol*, *12*(9), e1001940. doi:10.1371/journal.pbio.1001940
- Pandya, D. N., Petrides, M., Seltzer, B., & Cipolloni, B. P. (2015). *Cerebral cortex: Architecture, connections, and the dual origin concept.*: Oxford Press.
- Paquola, C., Bethlehem, R. A., Seidlitz, J., Wagstyl, K., Romero-Garcia, R., Whitaker, K. J., . . . Bullmore, E. T. (2019). Shifts in myeloarchitecture characterise adolescent development of cortical gradients. *elife*, *8*. doi:10.7554/eLife.50482
- Paquola, C., Vos De Wael, R., Wagstyl, K., Bethlehem, R. A. I., Hong, S. J., Seidlitz, J., . . . Bernhardt, B. C. (2019). Microstructural and functional gradients are increasingly dissociated in transmodal cortices. *PLoS Biol*, *17*(5), e3000284. doi:10.1371/journal.pbio.3000284
- Rilling, J. K., & Insel, T. R. (1999). Differential expansion of neural projection systems in primate brain evolution. *Neuroreport*, *10*(7), 1453-1459. doi:10.1097/00001756-199905140-00012
- Rinne, T., Muers, R. S., Salo, E., Slater, H., & Petkov, C. I. (2017). Functional Imaging of Audio-Visual Selective Attention in Monkeys and Humans: How do Lapses in Monkey Performance Affect Cross-Species Correspondences? *Cereb Cortex*, *27*(6), 3471-3484. doi:10.1093/cercor/bhx092
- Royer, J., & al., e. (biorXiv). <https://t.co/y4ZBG78WwH>.
- Salimi-Khorshidi, G., Douaud, G., Beckmann, C. F., Glasser, M. F., Griffanti, L., & Smith, S. M. (2014). Automatic denoising of functional MRI data: combining independent component analysis and hierarchical fusion of classifiers. *Neuroimage*, *90*, 449-468. doi:10.1016/j.neuroimage.2013.11.046
- Sanides, F. (1962). *Die Archtekonik des Menschlichen Stirnhirns*: Springer.
- Schaeffer, D. J., Hori, Y., Gilbert, K. M., Gati, J. S., Menon, R. S., & Everling, S. (2020). Divergence of rodent and primate medial frontal cortex functional connectivity. *Proc Natl Acad Sci U S A*, *117*(35), 21681-21689. doi:10.1073/pnas.2003181117
- Schleicher, A., Amunts, K., Geyer, S., Morosan, P., & Zilles, K. (1999). Observer-independent method for microstructural parcellation of cerebral cortex: A quantitative approach to cytoarchitectonics. *Neuroimage*, *9*(1), 165-177. doi:10.1006/nimg.1998.0385
- Schonwiesner, M., Dechent, P., Voit, D., Petkov, C. I., & Krumbholz, K. (2015). Parcellation of Human and Monkey Core Auditory Cortex with fMRI Pattern Classification and Objective Detection of Tonotopic Gradient Reversals. *Cereb Cortex*, *25*(10), 3278-3289. doi:10.1093/cercor/bhu124
- Stuber, C., Morawski, M., Schafer, A., Labadie, C., Wahnert, M., Leuze, C., . . . Turner, R. (2014). Myelin and iron concentration in the human brain: a quantitative study of MRI contrast. *Neuroimage*, *93 Pt 1*, 95-106. doi:10.1016/j.neuroimage.2014.02.026

- Taylor, C. M., Pritschet, L., Olsen, R. K., Layher, E., Santander, T., Grafton, S. T., & Jacobs, E. G. (2020). Progesterone shapes medial temporal lobe volume across the human menstrual cycle. *Neuroimage*, *220*, 117125. doi:10.1016/j.neuroimage.2020.117125
- Van Essen, D. C., Smith, S. M., Barch, D. M., Behrens, T. E., Yacoub, E., Ugurbil, K., & Consortium, W. U.-M. H. (2013). The WU-Minn Human Connectome Project: an overview. *Neuroimage*, *80*, 62-79. doi:10.1016/j.neuroimage.2013.05.041
- Worsley, K., Taylor, J. E., Carbonell, F., Chung, M. K., Duerden, E., Bernhardt, B. C., . . . Evans, A. (2009). SurfStat: A Matlab toolbox for the statistical analysis of univariate and multivariate surface and volumetric data using linear mixed effect models and random field theory. *Neuroimage*, *S102*.
- Xu, T., Nenning, K., Schwartz, E., Hong, S. J., Vogelstein, J. T., Fair, D. A., . . . Langs, G. (2020). Cross-species Functional Alignment Reveals Evolutionary Hierarchy Within the Connectome. *Neuroimage*.
- Xu, T., Sturgeon, D., Ramirez, J. S. B., Froudast-Walsh, S., Margulies, D. S., Schroeder, C. E., . . . Milham, M. P. (2019). Interindividual Variability of Functional Connectivity in Awake and Anesthetized Rhesus Macaque Monkeys. *Biol Psychiatry Cogn Neurosci Neuroimaging*, *4*(6), 543-553. doi:10.1016/j.bpsc.2019.02.005
- Zilles, K., Palomero-Gallagher, N., Grefkes, C., Scheperjans, F., Boy, C., Amunts, K., & Schleicher, A. (2002). Architectonics of the human cerebral cortex and transmitter receptor fingerprints: reconciling functional neuroanatomy and neurochemistry. *Eur Neuropsychopharmacol*, *12*(6), 587-599. doi:10.1016/s0924-977x(02)00108-6

REVIEWERS' COMMENTS

Reviewer #1 (Remarks to the Author):

The authors did a stellar job in addressing my comments. I have no further comments.

Reviewer #2 (Remarks to the Author):

The authors have made a very careful and exhaustive revision and addressed my concerns and those of the other reviewers. I personally think this work is very interesting, original and valuable to the literature.

Reviewer #3 (Remarks to the Author):

The authors should be commended for their thorough and fully responsive revision. I have no further comments and believe that the final version of the manuscript will undoubtedly have a large impact on the field.